# VUDG: A Dataset for Video Understanding Domain Generalization

**Ziyi Wang**[1,2]**, Zhi Gao**[1,2*]**, Boxuan Yu**[1]**, Zirui Dai**[1]**, Peiyao Wang**[1]**, Yuxiang Song**[1]**,
Qingyuan Lu**[1]**, Jin Chen**[1]**, Xinxiao Wu**[1,2]

[1]Beijing Key Laboratory of Intelligent Information Technology, School of Computer Science & Technology, Beijing Institute of Technology
[2]Guangdong Laboratory of Machine Perception and Intelligent Computing,  Shenzhen MSU-BIT University
`https://VUDG-Video.github.io`

## ABSTRACT

Video understanding has made remarkable progress in recent years, largely driven by advances in deep models and the availability of large-scale annotated datasets. However, the robustness of these models to domain shifts encountered in real-world video applications remains a critical yet underexplored problem, limiting their practical reliability. To address this problem, we introduce **V**ideo **U**nderstanding **D**omain **G**eneralization (**VUDG**), the first dataset designed specifically for evaluating domain generalization in video understanding. VUDG contains videos from 11 distinct domains that cover three types of domain shifts, and maintains semantic consistency across different domains to ensure fair and meaningful evaluation. We propose a multi-expert progressive annotation framework to efficiently annotate videos with structured question-answer pairs designed for domain generalization. Extensive experiments on 9 representative Large Vision-Language Models (LVLMs) and several traditional video question answering methods show that most models (including state-of-the-art LVLMs) suffer performance degradation under domain shifts. These results highlight the challenges posed by VUDG and the difference in the robustness of current models to data distribution shifts. We believe VUDG provides a critical resource to benefit future research in domain generalization for video understanding.

## 1 INTRODUCTION

Video understanding has achieved remarkable progress on tasks like action recognition and video question answering (VideoQA) (Chen & Ho, 2022; Lin et al., 2022; Li et al., 2022). However, most models assume the training and testing data have the same distribution, leading to significant performance degradation when encountering distribution shifts in real-world applications. This issue becomes more and more critical as large vision-language models (LVLMs) are increasingly fine-tuned for specific video applications. When deploying in the real world, a model's ability to handle unseen domains is essential for safety and reliability, as it is impossible to cover all potential data distributions during the fine-tuning phase. This problem could be formulated as the task of domain generalization (DG) (Lin et al., 2023b; Papadakis & Spyrou, 2024; Chen et al., 2023; Wang et al., 2026) in video understanding, where the model trained on the training data (source domain) is expected to perform well on the unseen testing data (target domain) with different distributions. Given the aforementioned challenges, evaluating the generalization of LVLMs after fine-tuning is an essential and valuable step to ensure their robustness in real-world scenarios.

Although there exist some benchmarks (Jang et al., 2017; Li et al., 2024a;b; Fu et al., 2024) of video understanding across different domains, they are not suitable to evaluate the DG performance, as their semantic spaces across domains are different, and the performances of models will be affected not only by the domain shifts but also by the semantic space differences. Hence, the robustness of models to the domain shifts, *i.e.,* the model's generalization, may not be evaluated properly. This

---

*Corresponding author: Zhi Gao.

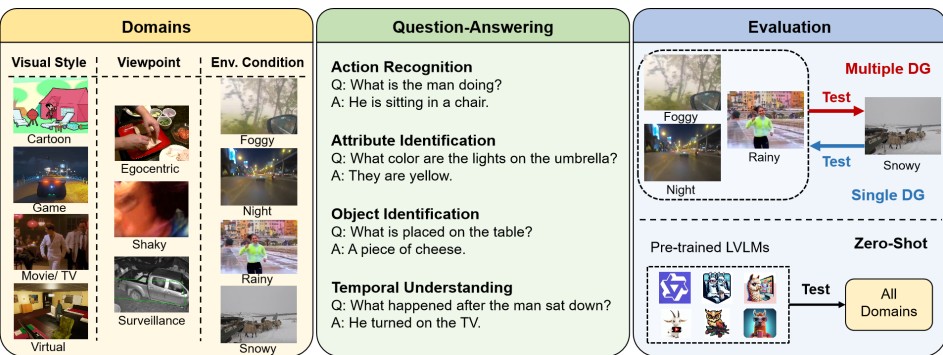

Figure 1: Overview of the proposed VUDG dataset.

highlights a critical gap: the lack of a dedicated dataset to rigorously and properly evaluate the domain generalization capabilities of video understanding models.

To bridge this critical gap, we formally identify and tackle the problem of domain generalization in video understanding. We introduce Video Understanding Domain Generalization (VUDG), the first dataset specifically designed to evaluate the performance of domain generalization in video understanding. VUDG comprises 11 domains, with videos sourced from diverse online platforms and open-source datasets (Wang et al., 2019; Ugai et al., 2024; Wang et al., 2024a; Chen et al., 2024). This collection strategy aims to maximize the diversity of videos while preserving inter-domain semantic consistency, a core requirement to evaluate the domain generalization properly. These domains exhibit variations in **visual styles** (*e.g.*, cartoon, game, movie/TV show, virtual environment), **viewpoints** (*e.g.*, egocentric, surveillance, shaky), and **environmental conditions** (*e.g.*, foggy, night, rainy, snowy), making VUDG a comprehensive generalization evaluation in video understanding. Note that we collect both the training and testing data for each domain, training and testing data are from distinct data sources, and the testing data is from testing splits of existing datasets or online platforms, to avoid the potential data leakage issue in evaluating LVLMs.

To construct VUDG, we introduce a progressive multi-expert annotation framework specifically designed for the DG task. A key feature of our framework is ensuring semantic consistency across all 11 domains by pre-defining a shared space of daily human activities. The framework then employs a cascade of distinct large models for automated QA generation and verification. This design mitigates self-reinforcement bias and streamlines the final human review process to ensure data quality. This process yields high-quality open-ended and multiple-choice QA pairs, making VUDG a versatile dataset to rigorously evaluate model generalization. The dataset is designed to support various DG protocols, including multi-source (Wang et al., 2021), single-source (Chen et al., 2023), and zero-shot (Lu et al., 2024) domain generalization.

We conduct extensive experiments on VUDG with representative VideoQA models and 9 state-of-the-art LVLMs under multi-source, single-source, and zero-shot DG settings. The results reveal that all models struggle to transfer into the target domains with unseen and different distributions, showing a significant performance drop compared to their performance on source domains. These experiments validate the challenges posed by VUDG and highlight the limitations of current models in handling domain shifts, motivating future work on developing more generalizable video understanding models.

In summary, our key contributions are as follows:

- We identify the problem of domain generalization in video understanding and introduce VUDG, the first dataset specifically designed to evaluate it. VUDG contains 11 different domains from varying visual styles, viewpoints, and environmental conditions, while sharing the same semantic space.

- We propose a progressive multi-expert annotation framework designed for DG that leverages a cascade of models for annotation generation and verification, minimizing model bias and human workload while ensuring high-quality open-ended and multiple-choice QA pairs.

- We conduct comprehensive experiments on 9 state-of-the-art LVLMs and several VideoQA baselines, revealing the challenge of VUDG on large gaps across domains and highlighting the limitations of current models under distribution shifts.

## 2 RELATED WORK

### 2.1 VIDEO UNDERSTANDING BENCHMARKS

The development of video understanding datasets has fueled recent advancements in video understanding. Datasets such as ActivityNet (Caba Heilbron et al., 2015) and Kinetics (Kay et al., 2017) have been pivotal in action recognition tasks, providing millions of labeled video clips across diverse activities. Later, Charades (Sigurdsson et al., 2016), TVQA (Lei et al., 2018), MSVD-QA (Xu et al., 2017), and MSR-VTT-QA (Xu et al., 2016) expand the scope of video understanding to include VideoQA tasks, introducing videos paired with textual questions that require reasoning about the video content. However, these datasets ignore scenarios where the distributions of the training and testing data are inconsistent, leaving the problem of domain generalization underexplored. Although some recent datasets (*e.g.*, VideoVista (Li et al., 2024b), Video-MME (Fu et al., 2024)) include videos from multiple categories such as HowTo, Film, and Cartoon, the semantic disparities between these categories are often too large. Therefore, they are not ideal for isolating the effects of domain shifts, making it difficult to fairly evaluate a model's generalization ability across domains in downstream tasks.

### 2.2 DOMAIN GENERALIZATION IN VIDEO TASKS

Previous works (Wang et al., 2024b; Zhang et al., 2023; Lin et al., 2023b; Yao et al., 2021) have explored domain generalization (DG) in video tasks to enhance robustness under unseen distributions. VideoDG (Yao et al., 2021) introduces an adversarial pyramid network and constructs three DG settings based on different dataset sources, different action consequences, and different camera viewpoints for video classification generalization. Ani-GIFs (Majumdar et al., 2022) presents the first synthetic DG dataset using animated GIFs and real videos to study domain shift in action recognition. ARGO1M (Plizzari et al., 2023) samples egocentric clips from Ego4D (Grauman et al., 2022) across diverse scenarios and locations to evaluate cross-context generalization. MDVAD (Flaborea et al., 2023) aggregates six surveillance video datasets to benchmark anomaly detection under environment and camera shifts. In contrast to these datasets that focus on domain generalization in video classification, anomaly detection, or action recognition, our dataset is tailored for video understanding, which poses richer visual reasoning challenges and aligns closely with the rapid development of LVLMs.

Table 1: The comparison of existing video understanding datasets involves several key aspects: total number of videos (Videos) and video clips (Clips), number of QA pairs (QA Pairs), annotation method (Anno., where M/A indicates manual/automatic), whether the videos contain diverse domains (Dom.), and whether the semantic spaces across domains are consistent (Sem.).

| Datasets | Videos | Clips | QA Pairs | Anno. | Dom. | Sem. |
|---|---|---|---|---|---|---|
| MSRVTT-QA (Xu et al., 2017) | 2,990 | 2,990 | 72,821 | A | ✗ | – |
| MSVD-QA (Xu et al., 2017) | 504 | 504 | 13,157 | A | ✗ | – |
| ActivityNet-QA (Yu et al., 2019) | 800 | 800 | 8,000 | M | ✗ | – |
| EgoSchema (Mangalam et al., 2023) | 5,063 | 5,063 | 5,063 | A&M | ✗ | – |
| TGIF-QA (Jang et al., 2017) | 9,575 | 9,575 | 8,506 | A&M | ✓ | ✗ |
| MVBench (Li et al., 2024a) | 3,641 | 3,641 | 4,000 | A | ✓ | ✗ |
| Video-Bench (Ning et al., 2023) | 5,917 | 5,917 | 17,036 | A&M | ✓ | ✗ |
| TempCompass (Liu et al., 2024) | 410 | 500 | 7,540 | A&M | ✓ | ✗ |
| Video-MME (Fu et al., 2024) | 900 | 900 | 2,700 | M | ✓ | ✗ |
| VideoVista (Li et al., 2024b) | 894 | 3,402 | 3,402 | A | ✓ | ✗ |
| VUDG (Ours) | 7,899 | 7,899 | 36,388 | A&M | ✓ | ✓ |

## 3 VUDG DATASET

We introduce the **VUDG** dataset that contains 11 distinct domains, including videos with different visual styles, viewpoints, and environmental conditions. To ensure high-quality data, we propose a progressive multi-expert annotation framework that leverages multiple large models, followed by human review for question-answer pairs generation and filtering. Importantly, we incorporate different large models in the generation and verification stages to mitigate the bias stemming from a fixed model that tends to validate its own outputs, thereby improving the diversity, objectivity, and robustness of the collected QA pairs. The annotation pipeline consists of four key stages: (a) **Video**

**Collection**, (b) **Open-Ended QA Pairs Generation**, (c) **Multiple-Choice QA Pairs Generation**, and (d) **QA Pairs Screening and Reviewing**. The overall workflow is illustrated in Figure 2.

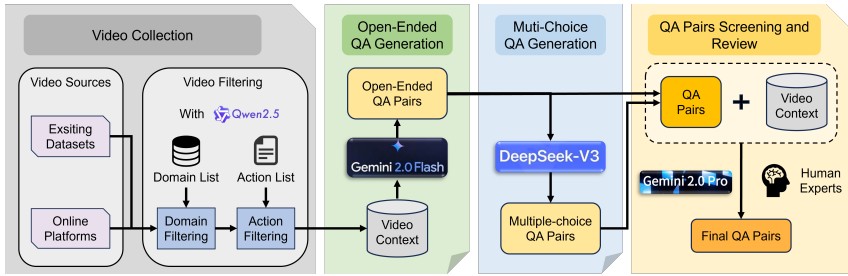

Figure 2: The pipeline diagram of the proposed multi-expert annotation framework.

## 3.1 VIDEO COLLECTION

We define 11 domains for video collection, including cartoon, game, movie/TV show, virtual environment, egocentric, surveillance, shaky, foggy, night, rainy, and snowy. Then, we collect videos from various data sources based on the domain name. We employ Qwen2.5-VL-7B (Bai et al., 2025) to filter out irrelevant videos that do not belong to the predefined domains. To ensure semantic consistency across domains, we manually define a list of daily human activity scenes (*e.g.*, reading books or documents, riding a bicycle, feeding a pet *etc.*) and utilize Qwen2.5-VL-7B to select videos belonging to this predefined activity list. The activity list and the prompts that are used to select videos are detailed in Appendix B.

To eliminate potential data leakage (since LVLMs are pre-trained on a large amount of data), we create training and testing sets for each domain and ensure a clear separation between them by collecting them from different data sources.

**Training Set:** The training set is constructed exclusively from the training sets of existing open-source datasets, including *InternVid* (Wang et al., 2024a), *ShareGPT4Video* (Chen et al., 2024), *VideoInstruct100K* (Maaz et al., 2024), and *MMDL* (Ugai et al., 2024). We have checked that no data from these sources overlaps with existing benchmarks (used as the testing set), ensuring strict separation between the training and testing data in VUDG.

**Testing Set:** The testing set is primarily derived from the testing sets of existing open-source video datasets, benchmarks, and videos crawled from online platforms. Specifically, we use the test splits of *VATEX* (Wang et al., 2019), *ActivityNet* (Caba Heilbron et al., 2015), VideoVista (Li et al., 2024b) and *MMDL* (Ugai et al., 2024). Furthermore, we leverage diverse user-generated content from online video platforms such as *YouTube*, *Douyin*, and *Bilibili*. These platforms host rich and varied videos from different domains, enabling us to collect a broad spectrum of videos. Table 2 shows the proportion of each source of the test set videos.

Table 2: Proportion of video sources in the test set.

| Datasets | VATEX | ActivityNet | MMDL | VideoVista | Self Collect |
|---|---|---|---|---|---|
| Proportion | 15.17% | 22.02% | 11.78% | 1.41% | **49.62%** |

All subsequent processes, such as QA pairs generation and filtering, are applied uniformly to all the collected videos, without further distinguishing between the training set and the testing set.

## 3.2 QUESTION AND ANSWER GENERATION

The question and answer generation process comprises three steps: open-ended QA pairs generation, multiple-choice QA pairs generation, and QA pairs screening and reviewing.

**Design of Question Category:** The generated QA pairs are expected to enable a thorough and comprehensive evaluation of video understanding. Figure 3 illustrates four types of questions: (1) **Action Recognition**, which focuses on identifying action categories, requiring the model to accurately

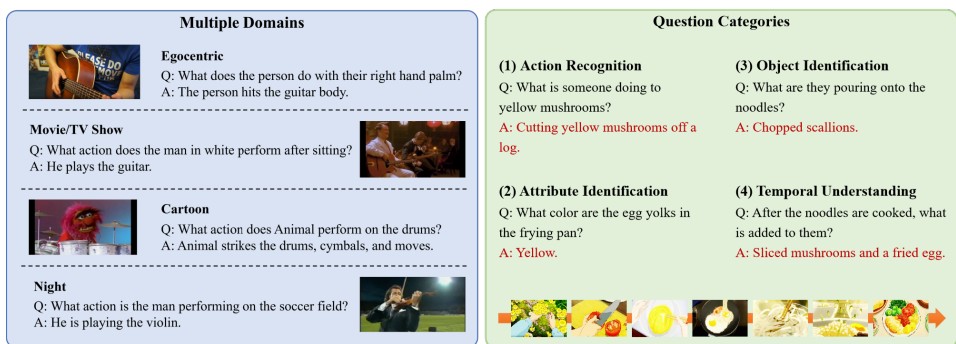

Figure 3: Overview of various domains and question types in our VUDG dataset.

recognize the action occurring at a specific time point in the video; (2) **Attribute Identification**, which assesses the model's ability to perceive visual attributes such as color, shape, and position of simple objects; (3) **Object Identification**, which tests the model's ability to recognize specific objects; (4) **Temporal Understanding**, which involves the temporal ordering of actions and requires the model to accurately identify the sequence of events, *i.e.,* the event that occurs before or after a specific action.

**Open-Ended QA Pairs Generation:** To generate open-ended QA pairs, we first leverage Gemini-2.5-Flash, a fast and cost-effective multimodal model with strong video understanding capabilities. This model is used to generate initial questions and open-ended answers for each video. We design two distinct prompts for question categories (1)–(3) and question category (4), respectively, since they focus on different information cues. Detailed examples of the two prompts can be found in Appendix J.1. For each video, we generate one question for each of the question categories (1)–(3) and two questions for the question categories (4).

**Multiple-Choice QA Pairs Generation:** For the generation of multiple-choice QA Pairs, we utilize DeepSeek-v3 to generate five plausible but incorrect options for each open-ended QA pair. These options are conditioned on the original question and the correct answer. Afterward, the options are randomized to ensure a balanced distribution of all six choices. Examples of prompts for multiple-choice QA generation can be found in Appendix J.2.

**QA Pairs Screening and Reviewing:** Despite the structured pipeline used for QA pair generation, issues such as ambiguous phrasing, semantically overlapping options, and factual inaccuracies in open-ended answers may still arise, potentially compromising the quality of the generated pairs. To address these challenges, we introduce a hybrid screening process that integrates both automated model-based evaluation and human-expert review. First, we employ Gemini-2.5-Pro, a more advanced multimodal model, to perform a thorough review of each QA pair with access to the original video context. This model classifies each QA pair into one of three categories: (a) correct QA pairs, (b) partially flawed answers with fixable issues, and (c) invalid questions. These automated classifications serve as the basis for further manual inspection. Then, human experts revise or remove QA pairs that are flagged as problematic ((b) or (c)) by Gemini-2.5-Pro, ensuring that the final dataset maintains high standards of clarity and accuracy. Crucially, this mandatory human-in-the-loop refinement, combined with our multi-model cascade strategy, effectively breaks the potential circularity loop and mitigates the risk of LLM dependence in dataset construction. Detailed prompts for Gemini-2.5-Pro can be found in Appendix J.3.

## 3.3 STATISTICS

The training set comprises 6,337 video clips and 31,685 QA pairs. The distribution of videos across domains is illustrated in Figure 4a. To reduce memory usage during training, all training videos are limited to a maximum duration of ten minutes. The duration distribution is shown in Figure 5a.

The testing set contains 1,532 video clips and 4,703 QA pairs, and the distribution of video numbers in each domain is demonstrated in Figure 4b. Compared to the training set, the testing set includes longer videos to better evaluate each model's ability to handle complex and extended temporal contexts. The duration distribution is shown in Figure 5b.

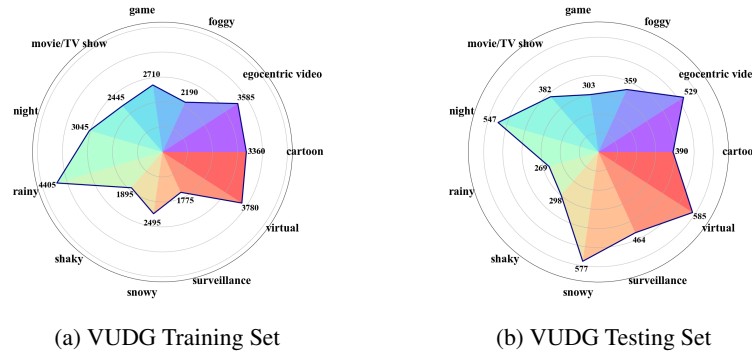

(a) VUDG Training Set         (b) VUDG Testing Set

Figure 4: Statistics showing the number of QA pairs across different domains in **(a)** the VUDG training set and **(b)** the VUDG testing set.

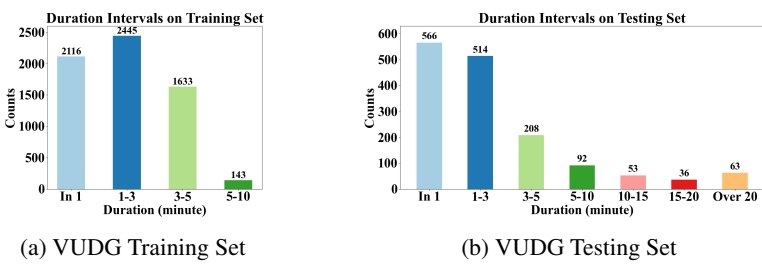

(a) VUDG Training Set         (b) VUDG Testing Set

Figure 5: Statistics illustrating the distribution of video durations across **(a)** VUDG training set and **(b)** VUDG testing set.

## 3.4 EVALUATION METRIC

We adopt two evaluation protocols widely used in domain generalization research: Leave-One-Domain-Out for multiple-domain generalization and Leave-But-One-Domain-Out for single-domain generalization, following prior works (Jo & Yoon, 2023; Chen et al., 2023).

For multiple domain generalization, one domain is used as the target domain using its testing set, while the training sets of the remaining $N-1$ domains are treated as source domains for training. The final performance is averaged over all domains, formulated as $\mathrm{Avg}^m = \frac{1}{N}\sum_{i=1}^{N} P_i$, where $P_i$ denotes the accuracy of the $i$-th domain used as the target domain, and $N$ is the number of domains in each setting.

For single domain generalization, the training set of one domain is used as the source domain for training, and the left $N-1$ domains are treated as target domains using their testing sets. The performance of this domain is calculated as the average of the test accuracy on the left $N-1$ domains, formulated as $\mathrm{Avg}^s = \frac{1}{N}\sum_{i=1}^{N}\left(\frac{1}{N-1}\sum_{j=1,j\neq i}^{N} P_j^i\right)$, where $P_j^i$ denotes the accuracy of the $j$-th domain with the $i$-th domain as the source domain.

For zero-shot generalization, models are directly tested on the full testing sets without any training. For multiple-choice questions, each model receives the video input alongside textual indications, including the question and the candidate options. We then compute the accuracy of a model on each domain and quantify the final performance by averaging the results on all domains. For open-ended questions, we use DeepSeek-V3 to automatically evaluate the answers. Specifically, we adopt task-specific evaluation protocols to evaluate from two aspects, with a total score of 5 points for each aspect and a maximum score of 10 for each question. For Q1, Q2 and Q3 (Action Recognition, Attribute Identification, and Object Identification), answers are evaluated based on factual accuracy and relevance to the question. For Q4 and Q5 (Temporal Understanding), which emphasize temporal reasoning, answers are assessed based on temporal accuracy and question relevance. The final score for a question is $\mathrm{Score} = S^{\mathrm{acc}} + S^{\mathrm{rel}}$, where $S^{\mathrm{acc}}, S^{\mathrm{rel}} \in [0,5]$. Evaluation prompts for all question types are detailed in Appendix J.4, while training configurations and model-specific hyperparameters for each evaluated model are included in Appendix E.

# 4 EXPERIMENTS

## 4.1 SETTINGS

**Baseline:** For domain generalization setting, we evaluate LLM-based and non-LLM-based methods. HBI (Jin et al., 2023) and EMCL4QA (Jin et al., 2022) are representative VideoQA methods that do not rely on LLMs, while VideoLLaMA2-7B (Cheng et al., 2024) and Qwen2.5VL-3B (Bai et al., 2025) are popular LVLMs using LLMs. We evaluate their performances on VUDG with the multiple and single domain generalization settings. For zero-shot evaluation, we evaluate nine large-scale video understanding models, including Video-ChatGPT-7B (Maaz et al., 2024), MiniGPT4-Video (Ataallah et al., 2024), VideoChat2-7B (Li et al., 2023), Video-LLaVA-7B (Lin et al., 2023a), VideoLLaMA2-7B (Cheng et al., 2024), mPLUG-Owl3-7B- (Ye et al., 2025), Video-CCAM-7B (Fei et al., 2024), VideoLLaMA3-7B (Zhang et al., 2025), Tarsier2-7B (Yuan et al., 2025) and Qwen2.5VL-7B (Bai et al., 2025).

**Implementation Details:** For training on the VUDG training set, we apply full-parameter fine-tuning to non-LLM-based methods and Low-Rank Adaptation (LoRA) (Hu et al., 2022) to LVLMs to ensure training efficiency. For LoRA, we set the rank to 128 and the scaling factor to 256. Further training details of each model can be found in Appendix E. For the zero-shot evaluation setting, we set all LVLMs to the official default configuration. Regarding models employing fixed frame sampling, we adopt their default official settings (*e.g.*, Video-LLaMA2 uses 16 frames per video). Regarding models evaluated under fixed FPS settings, we uniformly set the frame rate to 1 FPS to ensure consistency.

## 4.2 DOMAIN GENERALIZATION RESULTS

In the following experiments, we use abbreviations for different domains: **CA** (Cartoon), **GA** (Game), **MO** (Movie/TV), **VI** (Virtual), **EG** (Egocentric), **SU** (Surveillance), **SH** (Shaky), **FO** (Foggy), **NI** (Night), **RA** (Rainy), **SN** (Snowy), and **D-Avg** (Domain-wise Average). We first provide the fully finetuning results on all domains, which can serve as an upper bound for the performance of different methods on VUDG. The results are presented in Table 3

Table 3: Performance of full fine-tuning across all domains on multiple-choice QA.

| Model | Visual Style | | | | | Viewpoint | | | | Env. Condition | | | | | D-Avg |
|---|---|---|---|---|---|---|---|---|---|---|---|---|---|---|---|
| | CA | GA | MO | VI | Avg | EG | SU | SH | Avg | FO | NI | RA | SN | Avg | |
| VideoLLaMA2-7B (Cheng et al., 2024) | 65.4 | 61.4 | **72.5** | **71.1** | 67.6 | 74.7 | 68.8 | **68.5** | 70.6 | **69.4** | **69.7** | 62.8 | **73.1** | **68.7** | 68.8 |
| Qwen2.5VL-3B (Bai et al., 2025) | **71.8** | **62.1** | 71.5 | 70.8 | **69.0** | **76.2** | **69.2** | 66.1 | 70.5 | 66.3 | 67.6 | **65.8** | 70.2 | 67.5 | **68.9** |

We first evaluate HBI (Jin et al., 2023), EMCL4QA (Jin et al., 2022), VideoLLaMA2 (Cheng et al., 2024), and Qwen2.5VL (Bai et al., 2025) in single domain generalization setting under the domain shift caused by the difference of environment condition, *i.e.,* the Environmental Condition split of the VUDG dataset. As shown in Table 4, the performance of VideoLLaMA2 degrades 15.4 percentage points compared with that of full fine-tuning, indicating that single domain generalization poses a great challenge to these models.

Table 4: Single domain generalization results on multiple-choice QA under different domain shifts.

| Model | Visual Style | | | | | Viewpoint | | | | Env. Condition | | | | | D-Avg |
|---|---|---|---|---|---|---|---|---|---|---|---|---|---|---|---|
| | CA | GA | MO | VI | Avg$^s$ | EG | SU | SH | Avg$^s$ | FO | NI | RA | SN | Avg$^s$ | |
| EMCL4QA (Jin et al., 2022) | 18.4 | 16.7 | 18.5 | 17.9 | 17.9 | 19.0 | 17.4 | 16.4 | 17.6 | 18.8 | 18.2 | 17.1 | 16.8 | 17.7 | 17.7 |
| HBI(Jin et al., 2023) | 18.9 | 16.9 | 17.9 | 18.2 | 18.0 | 18.4 | 16.9 | 16.8 | 17.4 | 17.9 | 19.3 | 18.6 | 16.8 | 18.2 | 17.9 |
| VideoLLaMA2-7B (Cheng et al., 2024) | 53.2 | 48.7 | 44.9 | 47.8 | 48.6 | 52.4 | 54.4 | 52.5 | 53.1 | **63.1** | 56.6 | **60.4** | 53.0 | **58.3** | 53.4 |
| Qwen2.5VL-3B (Bai et al., 2025) | **63.3** | **66.5** | **66.1** | **64.6** | **65.1** | **59.8** | **61.7** | **63.4** | **61.6** | 57.2 | **58.4** | 57.3 | **58.4** | 57.8 | **61.5** |

Table 5 summarizes the performance of four VideoQA methods in multiple domain generalization setting. We test these methods under three types of domain shifts and record the average results of each type of domain shift. As shown in Table 5, we have several findings. Firstly, non-LLM-based methods (HBI (Jin et al., 2023) and EMCL4QA (Jin et al., 2022)) achieve poor performance, indicating their limited generalization ability when exposed to out-of-distribution domains. Secondly, LLM-based methods show significantly better generalization compared to non-LLM-based methods. Among them, VideoLLaMA2-7B outperforms all others with the highest average accuracy across Visual Style (66.5%), Viewpoint (66.2%), and Environmental Condition (68.2%). Qwen2.5VL-3B shows competitive results but suffers notable degradation under harsh environmental conditions (*e.g.*, only 55.9% on NI and 55.5% on SN), suggesting vulnerability to visual noise and degradation.

Despite VideoLLaMA2-7B's strong performance under multiple DG settings, a clear gap remains compared to models fine-tuned across all domains (Table 3). Meanwhile, Qwen2.5-VL-3B shows worse performance on multiple DG fine-tuning than its fully fine-tuning counterpart. These results suggest that current large vision-language models (LVLMs) require more robust training or fine-tuning strategies to enhance domain generalization when adapting to downstream tasks.

Table 5: Multiple domain generalization results on multiple-choice QA under different domain shifts.

| Model | Visual Style | | | | | Viewpoint | | | | Env. Condition | | | | | D-Avg |
|---|---|---|---|---|---|---|---|---|---|---|---|---|---|---|---|
| | CA | GA | MO | VI | $Avg^m$ | EG | SU | SH | $Avg^m$ | FO | NI | RA | SN | $Avg^m$ | |
| HBI (Jin et al., 2023) | 14.9 | 18.2 | 17.2 | 16.4 | 16.7 | 17.4 | 16.7 | 18.5 | 17.5 | 17.7 | 18.9 | 16.9 | 17.6 | 17.8 | 17.3 |
| EMCL4QA (Jin et al., 2022) | 17.7 | 18.7 | 16.8 | 17.7 | 17.7 | 17.4 | 16.7 | 19.6 | 17.9 | 19.1 | 18.4 | 18.8 | 18.3 | 18.7 | 18.1 |
| Qwen2.5VL-3B (Bai et al., 2025) | **70.5** | 60.7 | 62.0 | 66.2 | 65.0 | 65.6 | 61.2 | 57.7 | 61.5 | 61.0 | 55.9 | 59.5 | 55.5 | 58.0 | 61.4 |
| VideoLLaMA2-7B (Cheng et al., 2024) | 61.6 | **64.4** | **68.7** | **70.1** | **66.5** | **66.1** | **62.9** | **69.6** | **66.2** | **69.6** | **69.1** | **64.9** | **69.2** | **68.2** | **66.9** |

## 4.3 Zero-Shot Results

Table 6: Multiple-choice zero-shot test results on VUDG. Performance across 11 domains.

| Model | Visual Style | | | | | Viewpoint | | | | Env. Condition | | | | | D-Avg |
|---|---|---|---|---|---|---|---|---|---|---|---|---|---|---|---|
| | CA | GA | MO | VI | Avg | EG | SU | SH | Avg | FO | NI | RA | SN | Avg | |
| Video-ChatGPT-7B (Maaz et al., 2024) | 14.1 | 12.5 | 14.4 | 9.7 | 12.7 | 12.9 | 11.6 | 14.1 | 12.9 | 14.5 | 16.3 | 13.8 | 15.8 | 15.1 | 13.6 |
| MiniGPT4-Video (Ataallah et al., 2024) | 13.6 | 12.9 | 13.9 | 14.2 | 13.7 | 12.7 | 13.2 | 13.8 | 13.2 | 15.9 | 15.7 | 13.0 | 13.0 | 14.4 | 13.8 |
| VideoChat2-7B (Li et al., 2023) | 16.2 | 9.6 | 14.1 | 10.3 | 12.6 | 14.6 | 13.8 | 13.4 | 13.9 | 16.7 | 15.4 | 17.8 | 11.6 | 15.4 | 14.0 |
| Video-LLaVA-7B (Lin et al., 2023a) | 23.3 | 23.1 | 21.2 | 29.9 | 24.4 | 22.3 | 26.1 | 19.5 | 22.6 | 22.6 | 26.0 | 20.1 | 25.0 | 23.4 | 23.5 |
| VideoLLaMA2-7B (Cheng et al., 2024) | 31.5 | 34.0 | 31.7 | 30.4 | 31.9 | 34.6 | 34.5 | 39.6 | 36.2 | 33.4 | 34.7 | 30.5 | 32.2 | 32.7 | 33.4 |
| mPLUG-Owl3-7B (Ye et al., 2025) | 50.0 | 50.8 | 49.7 | 61.0 | 52.9 | 53.5 | 46.8 | 56.7 | 52.3 | 51.0 | 49.2 | 48.7 | 48.2 | 49.3 | 51.4 |
| Video-CCAM-7B (Fei et al., 2024) | 55.6 | 40.9 | 60.0 | 52.7 | 52.3 | 54.8 | 47.0 | 57.1 | 53.0 | 51.0 | 51.0 | 48.3 | 47.8 | 49.5 | 51.5 |
| VideoLLaMA3-7B (Zhang et al., 2025) | 69.7 | 63.7 | 67.3 | 74.0 | 68.7 | 66.0 | 58.4 | 61.1 | 61.8 | 64.6 | 63.1 | 64.3 | 64.1 | 64.0 | 65.1 |
| Tarsier2-7B (Yuan et al., 2025) | 64.6 | 56.8 | 59.7 | 75.4 | 64.1 | 66.0 | 64.0 | 63.8 | 64.6 | 57.7 | 63.4 | 59.5 | 60.5 | 60.3 | 62.8 |
| Qwen2.5VL-7B (Bai et al., 2025) | **71.3** | 61.4 | **72.3** | 75.4 | **70.1** | **79.8** | **73.3** | 68.1 | **73.7** | **77.2** | **69.7** | **71.0** | **73.7** | **72.9** | **72.1** |
| GPT-4o (16 frames) | 64.6 | **64.0** | 68.6 | 73.2 | 67.6 | 66.5 | 60.8 | **68.8** | 65.4 | 61.3 | 62.0 | 61.0 | 59.8 | 61.0 | 64.6 |

In this section, we report the zero-shot evaluation results on the testing set of VUDG. Domain-specific performance on multiple-choice QA pairs is presented in Table 6. We provide detailed results as well as domain-wise averages (D-Avg) across 11 domains. Table 7 presents the accuracy of multiple-choice questions categorized by question type.

**Multiple-Choice QA:** As shown in Table 6, Qwen2.5VL-7B achieves the highest average accuracy (72.1%) across 11 visual domains, demonstrating strong generalization. Meanwhile, the advanced closed-source model GPT-4o yields a lower accuracy of 64.6%, indicating that massive pre-training alone does not fully mitigate the challenge of domain shifts. Notably, models like Video-LLaMA3-7B show moderate performance but struggle on viewpoint shifts such as Surveillance (SU) and Shaky (SH) scenes. Earlier models like Video-ChatGPT-7B and MiniGPT4-Video exhibit much lower performance overall, suggesting limited robustness to various distribution shifts.

Table 7: Various question categories of multiple-choice test results on VUDG. **Q1** (Action Recognition), **Q2** (Attribute Identification), **Q3** (Object Identification), **Q4 & Q5** (Temporal Understanding).

| Model | Q1 | Q2 | Q3 | Q4 & Q5 | Overall |
|---|---|---|---|---|---|
| Video-ChatGPT-7B (Maaz et al., 2024) | 11.6 | 13.4 | 12.7 | 14.7 | 13.6 |
| VideoChat2-7B (Li et al., 2023) | 15.3 | 13.6 | 12.4 | 13.7 | 13.7 |
| MiniGPT4-Video (Ataallah et al., 2024) | 14.4 | 11.8 | 14.9 | 14.1 | 13.8 |
| Video-LLaVA-7B (Lin et al., 2023a) | 30.1 | 28.2 | 22.1 | 21.0 | 24.1 |
| VideoLLaMA2-7B (Cheng et al., 2024) | 35.3 | 36.2 | 36.3 | 30.2 | 33.3 |
| Video-CCAM-7B (Fei et al., 2024) | 53.8 | 55.0 | 58.0 | 47.1 | 51.5 |
| mPLUG-Owl3-7B (Ye et al., 2025) | 55.2 | 53.8 | 58.9 | 46.8 | 51.6 |
| VideoLLaMA3-7B (Zhang et al., 2025) | 64.3 | 75.2 | 73.7 | 59.0 | 65.4 |
| Qwen2.5VL-7B (Bai et al., 2025) | **73.7** | **77.3** | **80.8** | **67.7** | **72.7** |
| GPT-4o (16 frames) | 65.9 | 71.9 | 75.0 | 57.8 | 64.7 |

**Performance Across Question Categories:** To further understand model behavior, Table 7 breaks down performance by question category. Qwen2.5VL-7B achieves the best performance across all categories, with particularly strong results in Object Identification (Q3) and Temporal Understanding (Q4 & Q5). In addition, the performance of most models in action recognition, attribute identification, and object identification is generally better than that in temporal understanding. This pattern suggests that current LVLMs are better at static, appearance-based reasoning than dynamic temporal understanding, revealing a challenge for video-based generalization.

**Open-Ended QA:** We also evaluate the zero-shot performance of different methods on open-ended QA. As shown in Table 8, Video-CCAM achieves the highest domain average (6.84), followed closely by Qwen2.5VL-7B and mPLUG-Owl3-7B. Compared to the multiple-choice setting, performance

gaps among models narrow in the open-ended QA setting, suggesting increased difficulty in generating free-form answers under domain shifts.

Table 8: Open-ended zero-shot test results on VUDG. Performance across 11 visual domains.

| Model | Visual Style | | | | | Viewpoint | | | | Env. Condition | | | | | D-Avg |
|---|---|---|---|---|---|---|---|---|---|---|---|---|---|---|---|
| | CA | GA | MO | VI | Avg | EG | SU | SH | Avg | FO | NI | RA | SN | Avg | |
| MiniGPT4-Video (Ataallah et al., 2024) | 3.77 | 3.40 | 4.47 | 4.42 | 4.02 | 4.86 | 3.91 | 4.79 | 4.52 | 4.69 | 4.56 | 4.03 | 4.61 | 4.47 | 4.32 |
| Video-ChatGPT-7B (Maaz et al., 2024) | 5.34 | 5.10 | 5.63 | 5.63 | 5.43 | 5.88 | 5.36 | 5.87 | 5.70 | 5.70 | 5.81 | 5.31 | 5.87 | 5.67 | 5.59 |
| VideoChat2-7B (Li et al., 2023) | 5.22 | 5.33 | 5.71 | 5.82 | 5.52 | 6.12 | 5.44 | 5.98 | 5.85 | 5.77 | 5.96 | 5.17 | 5.88 | 5.70 | 5.67 |
| VideoLLaMA3-7B (Zhang et al., 2025) | 5.57 | 5.35 | 5.80 | 5.91 | 5.66 | 5.77 | 5.64 | 5.93 | 5.78 | 5.61 | 5.75 | 5.50 | 5.83 | 5.67 | 5.70 |
| Video-LLaVA-7B (Lin et al., 2023a) | 5.39 | 5.22 | 5.81 | 6.20 | 5.66 | 6.32 | 5.66 | 6.11 | 6.03 | 5.80 | 5.99 | 5.42 | 6.03 | 5.81 | 5.81 |
| VideoLLaMA2-7B (Cheng et al., 2024) | 5.80 | 5.91 | 6.32 | 6.68 | 6.18 | 6.88 | 6.05 | 6.65 | 6.53 | 6.46 | 6.55 | 6.12 | 6.45 | 6.40 | 6.35 |
| mPLUG-Owl3-7B (Ye et al., 2025) | 6.07 | 6.10 | 6.44 | 6.98 | 6.40 | 7.32 | 6.07 | 6.92 | 6.77 | 6.75 | 6.84 | 6.33 | 6.76 | 6.67 | 6.60 |
| Qwen2.5VL-7B (Bai et al., 2025) | **6.53** | 6.23 | **6.85** | 7.02 | 6.66 | **7.46** | 6.44 | 6.99 | 6.96 | **6.98** | 6.66 | **6.66** | **6.87** | 6.79 | 6.79 |
| Video-CCAM-7B (Fei et al., 2024) | 6.51 | **6.46** | 6.75 | **7.22** | **6.74** | 7.43 | **6.45** | **7.15** | **7.01** | 6.93 | **6.89** | 6.55 | **6.87** | **6.81** | **6.84** |

## 4.4 ANALYSIS ON TEXTUAL DOMAIN SHIFT

Beyond our primary focus on visual shifts, we conduct an auxiliary experiment to probe for textual domain shifts arising from different question formats. We evaluate VideoLLaMA2 on our multiple-choice (MC) test set under two fine-tuning conditions: using our open-ended (OE) training data versus our MC training data. As shown in Table 9, the model fine-tuned on OE data achieves 50.0% accuracy, a substantial 18.8 percentage point drop compared to the 68.8% from the model trained on in-domain MC data. This performance gap confirms the presence of a significant textual domain shift, demonstrating that linguistic variations alone can hinder model generalization.

## 4.5 HUMAN STUDY:

To validate our annotation quality, we conduct a human study. Since the multiple-choice QA pairs are generated from open-ended QA, we randomly sample 300 open-ended QA pairs and task 5 independent evaluators with rating them on a 5-point scale (1=Poor, 5=Excellent) for two key criteria: Relevance and Correctness. For comparison, we establish two control groups: (1) "random match" denotes randomly pairing questions with answers from other QA pairs, and (2) "filtered" represents QA pairs that were generated but subsequently discarded during the human review phase. As shown in Table 10, our annotations achieve substantially higher scores than both control groups, affirming the quality and reliability of the VUDG dataset.

Table 9: Performance of VideoLLaMA2 on the multiple-choice (MC) test set under different training conditions.

| Method | Accuracy (%) |
|---|---|
| VideoLLaMA2 (zero-shot) | 33.4 |
| VideoLLaMA2 (fine-tuned on OE) | 50.0 |
| VideoLLaMA2 (fine-tuned on MC) | 68.8 |

Table 10: Human evaluation of annotation quality.

| QA Pairs Type | Relevance | Correctness |
|---|---|---|
| Random-Matched | 1.23 | 1.45 |
| Filtered | 2.36 | 3.58 |
| Ours | 4.78 | 4.62 |

## 4.6 VISUALIZATION

We visualize the embeddings of video frames, questions, and ground-truth answers using CLIP-B/16 (Dosovitskiy et al., 2021) for image encoding and BGE-V1.5 (Xiao et al., 2024) for text encoding. As shown in Fig. 6, the CLIP frame features (a) exhibit relatively clear clustering by domain, indicating significant visual differences across domains and validating the challenge of domain shifts in VUDG. In contrast, the distributions of question embeddings (b) and answer embeddings (c) are more uniformly mixed, suggesting that the semantic content remains consistent across domains. This supports our design principle of preserving cross-domain semantic consistency while introducing realistic visual shifts.

## 5 CONCLUSIONS

We present VUDG, a novel dataset for evaluating domain generalization in video understanding while maintaining semantic consistency across 11 diverse domains, enabling fair and challenging evaluation. To construct high-quality QA pairs specifically for domain generalization, we develop a progressive multi-expert annotation pipeline that benefits from multiple large models together with human expert refinement. Through extensive experiments, we observe that current models struggle to address domain shifts when adapted to downstream tasks, resulting in suboptimal generalization. As for zero-shot evaluation, the performance of different LVLMs varies greatly, while even state-of-the-art LVLMs exhibit inconsistent performance between domains. We believe VUDG provides a valuable

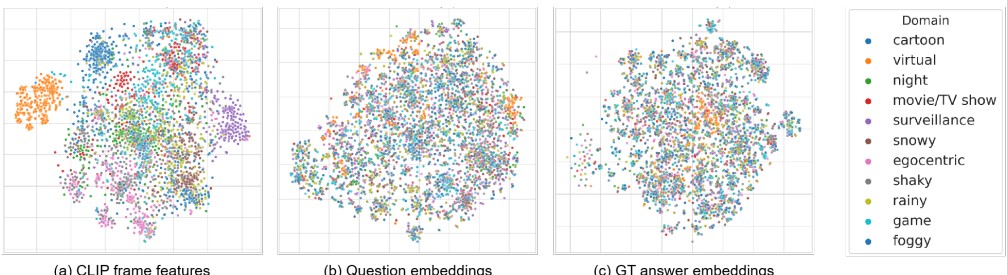

Figure 6: The t-SNE visualization of CLIP features for video frames, and BGE embeddings for questions and ground-truth answers across different domains.

foundation for advancing generalizable video understanding. However, our dataset currently only focuses on visual domain shifts. In the future, we plan to explore textual domain shifts and incorporate additional modalities such as audio to enable more comprehensive multimodal generalization.

## ACKNOWLEDGEMENTS

This work was supported by the Natural Science Foundation of China (NSFC) (Grant No. 62406009), the Shenzhen Science and Technology Program (Grant No. JCYJ20241202130548062), the Natural Science Foundation of Shenzhen (Grant No. JCYJ20230807142703006), the Guangdong Provincial Key Area Project of General Universities (Grant No. 2024ZDZX1017), and the Opening Project of the State Key Laboratory of General Artificial Intelligence, BIGAI/Peking University, Beijing, China (Grant No. SKLAGI2025OP03, SKLAGI2025OP08).

## REPRODUCIBILITY STATEMENT

The VUDG dataset and the codebase to reproduce all experiments are made publicly available.

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

## A    DOMAIN DEFINITIONS

The definitions of all domains are shown in Table 11.

Table 11: Domain definitions grouped by distribution shift types, along with descriptions.

| Shift Type | Domain | Description |
|---|---|---|
| Visual Style | Cartoon | Stylized animation with exaggerated motion and simplified texture |
| | Game | Scenes from video games, often rendered in real-time |
| | Movie/TV | Professionally shot narrative content with cinematic framing |
| | Virtual | Fully simulated environments, often from virtual production |
| Viewpoint | Egocentric | First-person perspective, often from head-mounted cameras |
| | Surveillance | Static, wide-angle views from mounted cameras in public/private spaces |
| | Shaky | Handheld, unstable footage with dynamic camera motion |
| Environmental Condition | Foggy | Reduced visibility due to simulated or natural fog |
| | Night | Low-light or nighttime scenes, often under artificial lighting |
| | Rainy | Outdoor scenes with visible rain, wet surfaces, and overcast skies |
| | Snowy | Scenes with snowfall, snow-covered ground, and diffused light |

## B    DATA COLLECTION

We present the prompt for Qwen2.5-VL-7B used to select videos that belong to the specific activity list in Figure 7 and the types of daily actions that the VUDG dataset focuses on are shown in Table 12.

---

**Prompts For Download**

"Below, I will provide you a set of phrases describing scenes of people's daily actions,
Please check if the video includes any scenes from the provided set.
If so, reply 'yes', otherwise answer 'no'. Do not answer any additional content."

---

Figure 7: Prompts for download.

The list of human daily activity scenes containing 37 items that the VUDG dataset focuses on is detailed in Table 12.

Table 12: Human Daily Activity Scenes List with Descriptions

| Activity | Description |
|---|---|
| Sitting | Sitting on a couch or chair |
| Using Laptop | Using a laptop or other electronic devices (including typing on a keyboard) |
| Writing/Drawing | Writing or drawing in a notebook (including painting) |
| Cooking | Cooking or preparing food (chopping, cutting, mixing, stirring) |
| Cleaning Dishes | Cleaning dishes or household items; organizing spaces |
| Gardening | Gardening or plant care (watering, tending, pruning) |
| Exercising | Exercising or stretching (yoga, home workouts, outdoor) |
| Hair Styling | Brushing or styling hair (combing, etc.) |
| Reading | Reading books or documents |
| Hygiene | Washing hands or doing hygiene routines (brushing teeth) |
| Meeting | Attending meetings or presentations (online or in-person) |
| Taking Notes | Taking notes during a meeting |
| Drinking | Drinking water, coffee, tea, or other beverages |
| Preparing Meals | Preparing a meal or snack (lunch, dinner) |
| Using Phone | Using a phone or tablet (checking notifications) |

| Activity | Description |
|---|---|
| Walking Outdoors | Walking in a park or natural area |
| Fixing Items | Adjusting or fixing household items |
| Feeding Pets | Feeding a pet |
| Art/Crafting | Engaging in art or crafting activities |
| Device Setup | Setting up or adjusting electronic devices |
| Checking Calendar | Checking or updating a calendar or planner |
| Watching Screens | Watching television or screens (incl. gaming) |
| Resting | Resting or lying down |
| Tool Preparation | Preparing tools or items for a task |
| Organizing | Cleaning or organizing a room (tidying clutter) |
| Laundry | Sorting laundry or folding clothes |
| Taking a Break | Taking a break from a task or activity |
| Conversation | Engaging in discussions or conversations |
| Shopping | Shopping or browsing items |
| Driving | Driving a car |
| Eating | Eating a meal |
| Playing Toys | Playing with a toy |
| Makeup | Applying makeup |
| Dancing | Dancing |
| Biking | Riding a bicycle |
| Music | Playing a musical instrument |
| Commuting | Commuting via public transport |

We also visualize the distribution of the number of QA pairs across different activity scenes in the test set in Figure 8.

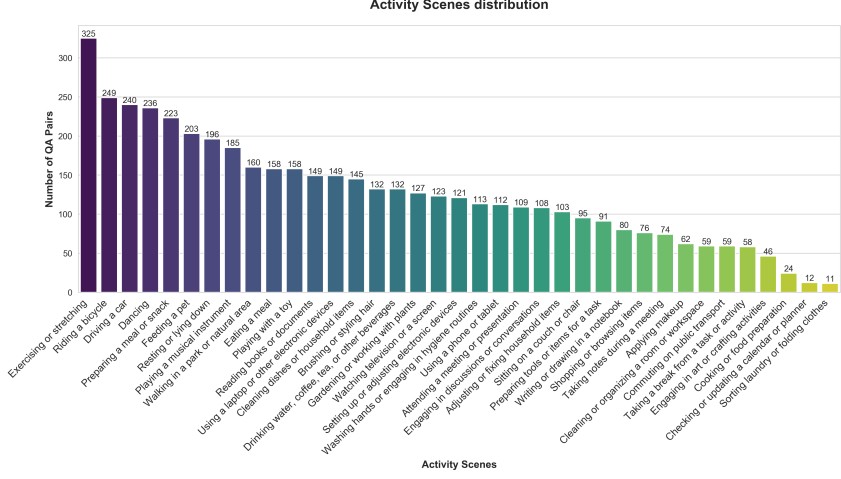

Figure 8: Distribution of the number of QA pairs across different activity scenes.

## C ANALYSIS ON STATIC BIAS

We conducted an in-depth analysis of language bias and visual bias in our constructed VUDG dataset, evaluating the Tarsier2-7B model under three input settings: text-only, one-frame sampling, and full-video input. Experimental results in Table 13 show that when relying solely on question text, the model achieves only 20.6% average accuracy, close to random guessing (theoretical baseline: 16.7%), indicating that the dataset contains minimal exploitable linguistic cues, effectively mitigating language bias. Meanwhile, performance under the one-frame setting (40.3%) is notably inferior to that of full-video input (62.8%), highlighting the critical role of temporal dynamics and contextual information

across video frames. This underscores the dataset's design emphasis on rich spatiotemporal reasoning and confirms its resistance to visual simplification bias.

Table 13: Multiple-choice zero-shot test results of **Tarsier2-7B** (Yuan et al., 2025) on VUDG under different settings: text-only, one-frame and full-video.

| Model | Visual Style | | | | | Viewpoint | | | | Env. Condition | | | | | D-Avg |
|---|---|---|---|---|---|---|---|---|---|---|---|---|---|---|---|
| | CA | GA | MO | VI | Avg | EG | SU | SH | Avg | FO | NI | RA | SN | Avg | |
| text-only | 23.3 | 21.5 | 20.7 | 18.3 | 20.9 | 21.0 | 22.6 | 22.8 | 22.1 | 20.6 | 20.3 | 17.8 | 17.9 | 19.2 | 20.6 |
| one-frame | 34.9 | 42.6 | 37.7 | 43.9 | 39.8 | 41.4 | 45.5 | 47.3 | 44.7 | 37.6 | 38.9 | 37.9 | 35.2 | 37.4 | 40.3 |
| full-video | 64.6 | 56.8 | 59.7 | 75.4 | 64.1 | 66.0 | 64.0 | 63.8 | 64.6 | 57.7 | 63.4 | 59.5 | 60.5 | 60.3 | 62.8 |

# D    SEMANTIC SPACE COMPARISON

As shown in Figure 9(b), Video-MME exhibits strong domain separation, particularly for "Sports Competition" and "Artistic Performance," which form distinct clusters far from other domains. This suggests the dataset encodes domain-specific biases that may encourage models to overfit to visual or stylistic cues rather than learn generalizable reasoning, making it less suitable for evaluating domain generalization. In contrast, VUDG (Figure 9(a)) maintains semantic consistency across domains, better reflecting real-world cross-domain video understanding challenges.

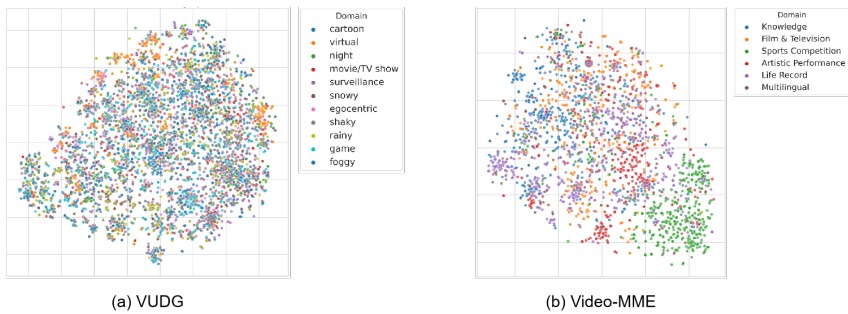

(a) VUDG                                       (b) Video-MME

Figure 9: Comparison of t-SNE visualization of BGE embedding for different domain problems between VUDG and Video-MME.

# E    IMPLEMENTATION DETAILS

All the experiments are conducted on 8 NVIDIA RTX 4090D GPUs, each with 24GB of graphics memory.

## E.1    HBI

We train HBI for 5 epochs with a global batch size of 32. On the model side, we cap captions at 32 words and videos at 12 frames, employ a 2D linear patch mode, set the slice-frame position to 2, keep all layers trainable (no freezing), and enable "loose" interaction modeling. Optimization is carried out with a base learning rate of 1e-4, a higher coefficient learning rate of 1e-3 for specialized submodules, and weighted loss terms of 2 for the KL divergence and 1 for the symmetric KL (SKL) divergence.

## E.2    EMCL4QA

We integrate the EMCL module into a CLIP-based video–question–answering backbone and train for 5 epochs with a global batch size of 128, clipping videos to 12 frames and questions to 32 tokens. During each forward pass, EMCL performs $T = 9$ routing iterations over $K = 32$ subspaces using a Gaussian kernel ($\sigma = 1$) and updates its bases via moving-average momentum $\alpha = 0.9$, then fuses reconstructed and original features with scale factor $\beta = 0.5$. We optimize end-to-end with Adam

and a 10% linear warmup, applying a learning rate of 1e-7 to the CLIP encoders and 1e-4 to the EMCL module and QA head, training against an InfoNCE-based cross-entropy loss with temperature $\tau = 0.01$.

### E.3  VIDEOLLAMA2

We fine-tune VideoLLaMA2 with a LoRA rank of $r = 128$ and an alpha of 256, inserting STC Connector modules as multimodal projectors learned at 2e-5. We train with a global batch size of 64 (per device 1, gradient accumulation 8), optimize via AdamW at 2e-5 with no weight decay and a 0.03 warmup ratio, and apply a cosine annealing schedule throughout. To enable efficient large-scale training, we leverage DeepSpeed with ZeRO Stage 3 optimization, which allows for memory-efficient distributed training without compromising performance.

### E.4  QWEN

We finetune Qwen2.5VL-3B using LoRA with a rank of 64 and a LoRA alpha of 64, targeting key projection modules ($q\_proj$, $k\_proj$, $v\_proj$, $o\_proj$) within the model architecture. We adopt a global batch size of 32, achieved by setting a per-device batch size of 4 and gradient accumulation steps of 4. Optimization is performed using the AdamW optimizer with a learning rate of 2e-7, no weight decay, and a warmup ratio of 0.03. A cosine learning rate scheduler is employed throughout training. To enable efficient large-scale training, we leverage DeepSpeed with ZeRO Stage 3 optimization, which allows for memory-efficient distributed training without compromising performance.

## F  THE USE OF LARGE LANGUAGE MODELS (LLMS)

Large Language Models (LLMs) were utilized as tools exclusively for dataset creation and evaluation, and not for research ideation. Our progressive multi-expert annotation framework employed a cascade of models for specific tasks: **Qwen2.5-VL-7B** was used for video selection from source datasets; **Gemini-2.5-Flash** generated initial open-ended question-answer (QA) pairs; **DeepSeek-v3** converted these into multiple-choice questions; and **Gemini-2.5-Pro** performed an automated quality check. This multi-model approach was adopted to mitigate the hallucination, bias issues, and the risk of circularity (i.e., evaluating models on data generated by models with similar architectures or relying solely on LLMs for benchmarking). Crucially, all machine-generated annotations underwent a final manual review and correction by human annotators to ensure high quality. For evaluation, we also used **DeepSeek-v3** as an automated judge to score the open-ended QA responses.

## G  ANALYSIS OF DOMAIN DISCREPANCY

To quantitatively verify that the domains within our dataset are sufficiently distinct, we measure the visual discrepancy between them using Jensen-Shannon (JS) divergence. We first extract frame-level visual features for videos in each domain using a pre-trained CLIP ViT-B/16 model. Then, for each pair of domains, we compute the JS divergence between their feature distributions. A higher JS divergence value indicates a larger gap and less similarity between the two domains.

Table 14 presents the results for several representative domain pairs that belong to the same high-level shift type. For instance, the game vs. virtual pair yields a high divergence of 0.441, and the egocentric vs. surveillance pair also shows a significant gap (0.333). These quantitative results confirm that our dataset contains meaningful and non-trivial domain shifts, providing a solid basis for evaluating domain generalization.

## H  SOCIETAL IMPACTS

Our work aims to improve the robustness and generalization of video-language models in real-world scenarios by introducing a dataset focused on domain generalization. This has potential positive societal impacts in enhancing the reliability of AI systems used in safety-critical or diverse environments, such as surveillance, autonomous vehicles, or assistive technologies, where domain shifts are inevitable.

Table 14: Jensen-Shannon (JS) divergence between feature distributions of representative domain pairs. Higher values indicate greater dissimilarity.

| Domain Pair | JS Divergence |
|---|---|
| game vs. cartoon | 0.312 |
| game vs. egocentric | 0.304 |
| game vs. surveillance | 0.289 |
| game vs. virtual | 0.441 |
| egocentric vs. surveillance | 0.333 |

Similar to many existing datasets, VUDG includes some publicly available videos collected from online platforms. While we take care to avoid personally identifiable or sensitive content, the use of web-sourced data may still raise concerns regarding content ownership or individual rights. To mitigate such risks, our dataset is released strictly for non-commercial, academic research purposes only. We encourage responsible use and further discussion on ethical data curation and model deployment practices in this field.

# I    QUESTION TYPES

In this section, we present illustrative examples for four question categories: Action Recognition (Table 15), Attribute Identification (Table 16), Object Identification (Table 17) and Temporal Understanding (Table 18).

Table 15: Examples of **Action Recognition** Questions.

| Question Type | No. | Question & Options | Answer |
|---|---|---|---|
| | 1 | **Q: What action does the character do upon exiting the door?** 
 A. The character waves at a friend. 
 B. The character jumps into a car. 
 C. The character runs down the street. 
 D. The character checks their watch. 
 **E. The character falls to their knees.** 
 F. The character starts singing loudly. | E |
| **Action Recognition** | 2 | **Q: What action does the person perform in the convenience store?** 
 A. pays for the items at the counter. 
 B. picks up a dropped item from the floor. 
 C. asks the clerk for assistance. 
 D. reads a magazine near the entrance. 
 E. drinks a soda near the fridge. 
 **F. puts items in the basket.** | F |
| | 3 | **Q: What action is the woman performing at the checkout counter?** 
 A. The woman is signing a receipt. 
 **B. The woman is talking to the cashier.** 
 C. The woman is packing her groceries. 
 D. The woman is searching for her wallet. 
 E. The woman is counting her change. 
 F. The woman is looking at her phone. | B |

Table 16: Examples of **Attribute Identification** Questions.

| Question Type | No. | Question & Options | Answer |
|---|---|---|---|
| | 1 | **Q: What is the relative position of the child and the ladder?**
A. The child is climbing up the ladder.
B. The child is sitting on top of the ladder.
**C. The child is standing below the ladder.**
D. The child is leaning against the ladder.
E. The child is holding the ladder steady.
F. The child is jumping off the ladder. | C |
| **Attribute Identification** | 2 | **Q: What is the quantity of full bookshelves behind the person?**
A. There are five full bookshelves.
B. There are three full bookshelves.
C. There is one full bookshelf.
D. There are four full bookshelves.
E. There are no full bookshelves.
**F. There are two full bookshelves.** | F |
| | 3 | **Q: What is the state of the fountains at the end of the video?**
A. The fountains are overflowing.
B. The fountains are frozen.
C. The fountains are turned off.
D. The fountains are being cleaned.
**E. The fountains are spraying water.**
F. The fountains are completely dry. | E |

Table 17: Examples of **Object Identification** Questions.

| Question Type | No. | Question & Options | Answer |
|---|---|---|---|
| | 1 | **Q: What is the tool attached to the end of the hose?**
A. It is a green nozzle.
B. It is a black brush.
C. It is a blue sprayer.
**D. It is a red baster.**
E. It is a white filter.
F. It is a yellow funnel. | D |
| **Object Identification** | 2 | **Q: What item is present near the stone henge area?**
A. A bucket is present near stone henge.
B. A flag is present near stone henge.
C. A rope is present near stone henge.
D. A mirror is present near stone henge.
**E. Bells are present near stone henge.**
F. A ladder is present near stone henge. | E |
| | 3 | **Q: What kind of artwork hangs on the left wall?**
A. a photograph of a mountain.
B. a print of a flower.
C. a drawing of a river.
D. a sketch of a tree.
E. a poster of a city.
**F. a painting of a tower.** | F |

Table 18: Examples of **Temporal Understanding** Questions.

| Question Type | No. | Question & Options | Answer |
|---|---|---|---|
| | 1 | **Q: What does father do right after entering the kitchen?**
A. He checks the time on the wall clock.
B. He washes his hands at the sink.
C. He opens the refrigerator to grab a drink.
D. He sits down at the kitchen table.
**E. He plays tricks on and kisses Mom.**
F. He starts cooking dinner immediately. | E |
| **Temporal Understanding** | 2 | **Q: What is the order of ingredients placed on the bread?**
A. Ham, lettuce, tomato, then cheese.
B. Lettuce, ham, tomato, then cheese.
C. Tomato, ham, lettuce, then cheese.
D. Cheese, ham, tomato, then lettuce.
**E. Lettuce, tomato, ham, then cheese.**
F. Tomato, lettuce, ham, then cheese. | E |
| | 3 | **Q: In what order does the main character demonstrate biking surfaces?**
**A. Clean pavement → Icy pavement → Snowy path.**
B. Snowy path → Clean pavement → Icy pavement.
C. Snowy path → Icy pavement → Clean pavement.
D. Clean pavement → Snowy path → Icy pavement.
E. Icy pavement → Clean pavement → Snowy path.
F. Icy pavement → Snowy path → Clean pavement. | A |

## J PROMPTS FOR GENERATION AND EVALUATION

### J.1 PROMPTS FOR OPEN-END QA PAIRS GENERATION

Prompts for open-ended QA pairs generation are shown in Figure 10 and Figure 11.

### J.2 PROMPTS FOR MULTIPLE-CHOICE OPTIONS GENERATION

Prompts for multiple-choice options generation are shown in Figure 12 and Figure 13.

### J.3 PROMPT FOR REVIEWING QA PAIRS

Prompts for reviewing QA pairs using Gemini-2.5-Pro are shown in Figure 14 and Figure 15.

### J.4 PROMPTS FOR OPEN-ENDED EVALUATION

Prompts for open-ended evaluation using DeepSeek-V3 are shown in Figure 16 and Figure 17.

## Prompts For Open-ended generation Q123

Please generate three open-ended questions and their answers based on the following video content. Follow the requirements below:

       1. The first question should be related to an action in the video. It should ask about a specific action in the video.
       2. The second question should be related to the attributes of objects in the video, such as their color, size, shape, position, distance, quantity, motion, material, state, orientation, or texture.
       3. The third question should be related to an object in the video. It should ask about the identification, description, or role of a specific object in the video.
       4. Each question should be precise and unambiguous, ensuring that the answer is clear and directly related to the video content.
       5. IMPORTANT: The questions and answers must be based solely on the visual content of the video. Do not use any information from the audio track, even if it is present in the video.
       6. Don't generate any other content except three QA pairs.
       7. Make your answers as simple as possible, try to use no more than 10 words.

Output Format:
Q1: [Question about action]
A1: [Answer to Q1]
Q2: [Question about attributes]
A2: [Answer to Q2]
Q3: [Question about object]
A3: [Answer to Q3]

Example Output:
Q1: Which did the man perform in the cinema?
A1: He ate popcorns.
Q2: What is the color of the woman's hat?
A2: Brown.
Q3: What is in the man's hand?
A3: An ice cream.

Figure 10: Prompts for generating open-ended answers for Q1, Q2 and Q3.

**Prompts For Open-ended generation Q45**

Please generate two open-ended questions and their answers based on the following video content. Follow the requirements below:

    1. Both questions must be related to temporal information or sequential relationships in the video, such as the order in which events occur or the sequence in which objects appear or reasoning about temporal information.
    2. Each question should be precise and unambiguous, ensuring that the answer is clear and directly related to the video content.
    3. IMPORTANT: The questions and answers must be based solely on the visual content of the video. Do not use any information from the audio track, even if it is present in the video.
    4. Don't generate any other content except two open-ended QA pairs.
    5. Make your answers as simple as possible while keeping it correct and precise, try to use no more than 10 words.

    Output Format:
    Q1: [Question about temporal or sequential information]
    A1: [Answer to Q1]
    Q2: [Question about temporal or sequential information]
    A2: [Answer to Q2]

    Example Output:
    Q1: Which event happens first for the characters in the video?
    A1: Waking up from bed.
    Q2: What is the correct order of events in the video?
    A2: Event A → Event B → Event C.

Figure 11: Prompts for generating open-ended answers for Q4 and Q5.

**Prompts For Options Generation Q123**

Based on the following questions and their open-ended answers, generate four multiple-choice options for each question. Follow the requirements below:

    1. For each question, generate six options (A, B, C, D, E, F), with only one correct option that matches the open-ended answer exactly.
    2. The other five options should be relevant to the question but significant different from the open-ended answer, and should not be ambiguous or confusing.
    3. The correct option should be marked with the corresponding letter (e.g., "A)").
    4. Ensure that the length of all six options is as similar as possible. Avoid making the correct option the longest one. Remember to simplify the content in the correct option to make it shorter without changing the meaning.
    5. Ensure that the semantic differences between the options are significant. Avoid creating options that are similar in their attributions, such as color, size, applications, or actions, objects etc.
    6. The output should be in the following format:
    Q1: [Question 1]
    A) [Option A]
    B) [Option B]
    C) [Option C]
    D) [Option D]
    E) [Option E]
    F) [Option F]
    Answer: [Correct Option Letter]

    Q2: [Question 2]
    A) [Option A]
    B) [Option B]
    C) [Option C]
    D) [Option D]
    E) [Option E]
    F) [Option F]
    Answer: [Correct Option Letter]

    Q3: [Question 3]
    A) [Option A]
    B) [Option B]
    C) [Option C]
    D) [Option D]
    E) [Option E]
    F) [Option F]
    Answer: [Correct Option Letter]

Example:
Q1: What action does the person perform immediately after entering the room?
    A) sits on the chair.
    B) picks up a book from the table.
    C) turns on the TV.
    D) opens the window.
    E) stands on the sofa.
    F) jumps on the desk.
    Answer: D

Q2: What is the color of the woman's hat?
    A) red.
    B) yellow.
    C) blue.
    D) pink.
    E) orange.
    F) cyan.
    Answer: B

Q3: What object is the person holding during the entire scene?
    A) a phone.
    B) a coffee mug.
    C) a book.
    D) a pen.
    E) a pencil.
    F) a hat.
    Answer: C

Questions and Answers:
Q1: {questions[0]}
A1: {open_answers[0]}
Q2: {questions[1]}
A2: {open_answers[1]}
Q3: {questions[2]}
A3: {open_answers[2]}

Figure 12: Prompts for generating options for Q1, Q2 and Q3.

**Prompts For Options Generation Q45**

Based on the following questions and their open-ended answers, generate four multiple-choice options for each question. Follow the requirements below:

1. For each question, generate six options (A, B, C, D, E, F), with only one correct option that matches the open-ended answer exactly.

2. The other five options should be relevant to the problem but significant different from the open-ended answer, especially in timing, and should not be ambiguous or confusing.

3. Ensure the options are clear, concise, and directly related to the question, but the temporal sequence of four options should not be similar.

4. The correct option should be marked with the corresponding letter (e.g., "A)").

5. Ensure that the length of all six options is as similar as possible. Avoid making the correct option the longest one. Remember to simplify the content in the correct option to make it shorter without changing the meaning.

6. Ensure that the semantic differences between the options are significant. Avoid creating options that are similar in timing. You may need to avoid temporal similarities between options.

7. The output should be in the following format:
    Q1: [Question 1]
    A) [Option A]
    B) [Option B]
    C) [Option C]
    D) [Option D]
    E) [Option E]
    F) [Option F]
    Answer: [Correct Option Letter]

Q2: [Question 2]
    A) [Option A]
    B) [Option B]
    C) [Option C]
    D) [Option D]
    E) [Option E]
    F) [Option F]
    Answer: [Correct Option Letter]

Example:
Q1: What action does the person perform immediately after entering the room?
    A) sits on the chair.
    B) picks up a book from the table.
    C) turns on the TV.
    D) opens the window.
    E) stands on the sofa.
    F) jumps on the desk.
    Answer: D

Q2: What is the color of the woman's hat?
    A) red.
    B) yellow.
    C) blue.
    D) pink.
    E) orange.
    F) cyan.
    Answer: B

Questions and Answers:
Q1: {questions[0]}
A1: {open_answers[0]}
Q2: {questions[1]}
A2: {open_answers[1]}

Figure 13: Prompts for generating options for Q4 and Q5.

## Prompts For Checking Q123

You are a strict video content auditor. Carefully examine each QA pair below based strictly on the visual information in the video.

Evaluation Criteria (MUST check all aspects):
1. Action Correctness: Are described actions consistent with the video?
2. Attribute Accuracy: Are object attributes (color/size/position) accurate?
3. Object Presence: Do mentioned objects actually exist in the video?
4. Action Sequence: Is the order of actions correctly described?
5. Temporal Consistency: Does the timing match the video progression?

Validation Rules:
- If BOTH Q and A are fully correct → Output TRUE
- If Q is factually wrong (asks about non-existent content) → Output FALSE
- If Q is valid but A is incorrect → Provide corrected answer
- If uncertain due to ambiguous video → Output FALSE

Q1: {q1}
A1: {a1open}

Q2: {q2}
A2: {a2open}

Q3: {q3}
A3: {a3open}

Required Output Format (STRICT):
A1result: TRUE / FALSE / [Corrected Answer]
A2result: TRUE / FALSE / [Corrected Answer]
A3result: TRUE / FALSE / [Corrected Answer]

Example output:
A1result: TRUE
A2result: FALSE
A3result: The Sky should be blue.

Critical Requirements:
1. ONLY use visual evidence from the video frames, do not consider any audio information.
2. Be EXTREMELY strict - assume FALSE when uncertain
3. For corrections: do not point out the original error, but provide the correct answer directly
4. Never explain your reasoning

Figure 14: Prompts for Checking Q1, Q2 and Q3.

## Prompts For Checking Q45

You are a strict video content auditor. Carefully examine each QA pair below based strictly on the visual information in the video.

Evaluation Criteria (MUST check all aspects):
1. Action Correctness: Are described actions consistent with the video?
2. Attribute Accuracy: Are object attributes (color/size/position) accurate?
3. Object Presence: Do mentioned objects actually exist in the video?
4. Action Sequence: Is the order of actions correctly described?
5. Temporal Consistency: Does the timing match the video progression?

Validation Rules:
- If BOTH Q and A are fully correct → Output TRUE
- If Q is factually wrong (asks about non-existent content) → Output FALSE
- If Q is valid but A is incorrect → Provide corrected answer
- If uncertain due to ambiguous video → Output FALSE

Q1: {q1}
A1: {a1open}

Q2: {q2}
A2: {a2open}

Required Output Format (STRICT):
A1result: TRUE / FALSE / [Corrected Answer]
A2result: TRUE / FALSE / [Corrected Answer]

Example output:
A1result: TRUE
A2result: FALSE

Critical Requirements:
1. ONLY use visual evidence from the video frames, do not consider any audio information.
2. Be EXTREMELY strict - assume FALSE when uncertain
3. For corrections: do not point out the original error, but provide the correct answer directly
4. Pay more attention on Temporal Consistency.
5. Never explain your reasoning

Figure 15: Prompts for Checking Q4 and Q5.

**Prompts for Open-ended Evaluation of Q123**

Evaluate the answer on TWO dimensions: Accuracy and Relevance.
Each score must be an integer from 0 to 5.
1) Accuracy:
   0: Completely incorrect; contradicts the video's content.
   1: Mostly incorrect; only minor correct elements.
   2: Some correct elements; significant inaccuracies present.
   3: Generally correct; minor inaccuracies.
   4: Mostly correct; very minor errors.
   5: Completely accurate; aligns perfectly with the video's content.
2) Relevance:
   0: Entirely irrelevant; does not address the question.
   1: Minimally relevant; largely off-topic.
   2: Partially relevant; includes significant unrelated information.
   3: Mostly relevant; some extraneous details.
   4: Highly relevant; minor non-essential details.
   5: Fully relevant; directly addresses the question without deviation.
Instructions:
   1. Analyze the respondent's answer in the context of the video. Scores need to be given with
      sufficient reasons.
   2. Assign accuracy_score (0-5) based on the correctness of the information.
   3. Assign relevance_score (0-5) based on how well the answer addresses the question.
   4. The answer's phrasing does not need to match the expected wording exactly—meaning
      and content alignment are what matter.
Output Format (JSON):
```json
{{
  "accuracy_score": <int 0-5>,
  "relevance_score": <int 0-5>
}}
```

Example:
Question: What is someone doing with a knife to yellow mushrooms?
Ground Truth: A person is shown using a knife to cut yellow mushrooms off a log in the
forest
Respondent's Answer: Someone is cutting up yellow mushrooms with a knife.
Expected JSON Output:
Output Format (JSON):
```json
{{
"accuracy_score": 4,
"relevance_score": 5
}}
```

Figure 16: Prompts for Open-ended Evaluation of Q1 to Q3(Action Recognition, Attribute Identification, and Object Identification)

## Prompts for Open-ended Evaluation of Q45

Evaluate the answer on TWO dimensions: Sequence Correctness and Relevance.
Each score must be an integer from 0 to 5.
---
1) Sequence Correctness:
   0: Completely incorrect order.
   1: Mostly incorrect order.
   2: Some correct order; significant errors.
   3: Generally correct order; minor errors.
   4: Mostly correct order; very minor errors.
   5: Perfect sequence.
2) Relevance:
   0: Entirely irrelevant; does not address the question.
   1: Minimally relevant; largely off-topic.
   2: Partially relevant; includes significant unrelated information.
   3: Mostly relevant; some extraneous details.
   4: Highly relevant; minor non-essential details.
   5: Fully relevant; directly addresses the question without deviation.
Instructions:
   1. Analyze the respondent's answer in the context of the video; provide brief
      justification for each score.
   2. Assign sequence_correctness_score (0-5) based on how accurately the answer
      reflects the temporal order of events.
   3. Assign relevance_score (0-5) based on how directly the answer addresses the
      question's focus.
   4. The answer's phrasing does not need to match the expected wording exactly—
      meaning and order alignment are what matter.
Output Format (JSON):
```json
{{
"sequence_correctness_score": <int 0-5>,
"relevance_score": <int 0-5>
}}
```

Example:
Question: In what order do the girl's expressions change?
Ground Truth: The video shows a girl listening through a safe, looking confused, then
later happily taking a shower.
Respondent's Answer: The girl's expression is confused as she listens through the safe,
then she is happy when she takes a shower.
Model Answer: Surprise, sadness, and then happiness.
Expected JSON Output:
```json
{{
 "sequence_correctness_score": 2,
 "relevance_score": 5
}}
```

Figure 17: Prompts for Open-ended Evaluation of Q4 and Q5(Temporal Understanding)

