# OpenReview forum: "VUDG: A Dataset for Video Understanding Domain Generalization"
_ICLR.cc/2026/Conference — ICLR 2026 Poster_

### Official Review · Reviewer_Qtow · 2025-10-31

**Soundness:** 2
**Presentation:** 3
**Contribution:** 2
**Rating:** 2
**Confidence:** 4

**Summary:**

This paper proposes a new dataset, VUDG, to test domain generalisation in video understanding models. It includes 11 domains across three kinds of shifts: **semantic** (e.g., cartoon), **viewpoint** (e.g., egocentric), and **environmental conditions** (e.g., foggy). VUDG has the following qualities:
1. There is semantic consistency across the domains, i.e., they show similar actions.
2. The annotation is based on a generating QA pairs from a cascade of LLMs with a final human review component.
3. The videos are sourced from existing datasets as well as from online video sites.

A suite of video models are evaluated on VUDG across 4 settings:
1. Full domain (train on all domains, test on all domains)
2. Multi-domain (train on all-but-one domain, test on the one remaining domain)
3. Single-domain (train on a single domain, test on all other domains)
4. Zero-shot

Some key findings include:
* Video models perform worse in settings (2) and (3) compared to (1).
* LLM based models are much superior than traditional domain generalisation methods.
* Qwen2.5VL-7B is very strong in zero-shot performance

**Strengths:**

1. Domain generalisation is an interesting problem to study in video context (especially for something like exo-to-ego generalisation).
2. The use of a cascade of strong LLMs to obtain QA annotations seems scalable and can possibly help scale up the proposed dataset. Including newer LLMs can also improve the current dataset.
3. The inclusion of "Temporal understanding" in the designed questions (e.g., related to temporal ordering of actions) is valuable and worth studying in context of domain generalisation.

**Weaknesses:**

1. **General concern about relevance of domain generalisation in LLM era**. Given the increasingly massive sizes and extent of the training datasets for modern video LLMs (e.g., Tarsier 2 is trained on 40 million videos), the gap between what we deem as training and testing is shrinking. This is likely why Qwen2.5VL-7B zero-shot (~72%) is stronger than any fine-tuned models. I am sure the closed models are even stronger just zero-shot. Thus, I am not sure if studying domain generalisation (and this dataset in particular) brings a lot of novel insights.

2. **Some stronger models can be included in evaluation.** An open model like Tarsier/Tarsier2 is shown to be quite a strong video LLM. It may already achieve very strong zero-shot performance, so it should be included in my view. Likewise, no closed models are included. While I understand the cost aspect, at least one model should be included because I suspect it may already be very strong zero-shot.

3. **Value added by this dataset.** While domain generalisation in video context sounds interesting, I did not gather what new, surprising insights this dataset would bring to the community. Studying it for special applications like medical imaging or private videos is valueable, but this dataset still has domains that are likely present in the train dataset of some of these big models.

**Questions:**

1. Domain generalisation is well studied for images. What about videos is special that makes it interesting in this context? If the authors can motivate this better, it can be useful.

2. it is not clear how you get data for different domains. The source datasets are stated but how does one get videos from different domains?

3. "Attribute identification" task may be more relevant for images than videos. Can you provide examples for attribute identification from videos?

4. Most existing video benchmarks suffer from static biases (a single frame or even without any visual frames, models can get reasonable accuracies). Have you done any analysis in this regard?

---

> ### Author Response · Authors · 2025-11-21
>
> **1. Regard General concern about relevance of domain generalisation in LLM era.**
>
> We greatly appreciate the insightful question raised by the reviewers. We agree that large-scale pre-training has indeed significantly improved the zero-shot generalization capabilities of modern LVLMs. However, we believe this precisely highlights the necessity of domain generalization benchmarks like VUDG. First, the best-performing Qwen2.5VL model achieves 72.1% on VUDG, indicating significant room for improvement. Second, training on massive, often proprietary datasets (such as 40 million videos) is difficult to achieve in terms of computational resources, cost, and data accessibility for the broad research community. Furthermore, many real-world applications, particularly those deployed on edge devices or requiring low latency, necessitate the use of smaller, more efficient models. In these scenarios, models still need to be trained or fine-tuned on limited domain data, making domain generalization a long-standing challenge that must be addressed. Since video understanding domain generalization is a fundamental problem, our dataset provides a first and feasible platform to fully evaluate the performance of video understanding domain generalization algorithms, not just the zero-shot generalization ability of large models.
>
> **2. Regard Adding Large Pre-trained Model**
>
> We incorporated Tarsier2-7B and GPT-4o (gpt-4o-2024-11-20) to enhance comprehensiveness. Tarsier2 achieves SOTA on the Virtual domain but lags in others (e.g., Game, Foggy), highlighting that massive pre-training does not guarantee universal robustness. For GPT-4o (16-frame sampling), accuracy was 64.6%, lower than Qwen2.5VL-7B and a fully fine-tuned VideoLLaMA2 (68.8%). These results confirm that both fine-tuned and large closed-source models struggle with domain shifts, validating VUDG's challenge. We have added these details to the revision.
>
> **3. Regard Common Domain**
>
> Although some domains exist in large models' pre-training corpora, the performance degradation of large pre-trained models (Tarsier2, GPT-4o) on VUDG proves that pre-training does not guarantee good domain generalization, necessitating fine-tuning. VUDG's value lies not in introducing rare domains, but in providing a systematic benchmark to measure this gap. Additionally, we deliberately selected these domains to maintain cross-domain semantic consistency, thus enabling a controlled evaluation of intrinsic generalization, which is crucial for developing robust architectures and generalized algorithms.
>
> **4. Regard Difference Between Image and Video DG**
>
> While image DG is well-studied, video DG introduces unique challenges stemming from the temporal dimension. First, appearance shifts (e.g., fog) compound difficulty by obscuring motion cues, impacting temporal reasoning even if the action is unchanged. Second, videos possess purely temporal domain shifts such as variations in action speed, camera stability, and rhythm, which have no parallel in static images. VUDG is explicitly designed to evaluate models' robustness against both appearance-based and temporal shifts.
>
> **5. Regard Way to Get Data for Different Domains**
>
> We first pre-defined 11 distinct domains (e.g., cartoon, egocentric, foggy) and collected videos using domain-specific keywords. To ensure purity, we then filtered content using Qwen2.5VL-7B followed by a rapid human inspection to verify domain relevance. As we have detailed in Section 3.1, this rigorous process ensured a clean, well-defined dataset for each domain.
>
> **6. Regard Attribute Identification**
>
> Attribute recognition in video presents unique challenges due to temporal dynamics. Many questions require localizing specific moments before judgment (e.g., a person's expression at a climax), which cannot be answered from a random frame. Furthermore, some attribution identification questions explicitly demand temporal understanding, such as "What is the state of the fountains at the end?". We have added representative examples in Section I of the Appendix.
>
> **7. Regard Static Biases**
>
> We hope to address this question from two aspect. First, to analyze language bias, we tested Tarsier2 with text-only input; it achieved 20.6% accuracy (near the 16.7% random baseline), proving strong reliance on visual content. Second, to address visual bias, we have explicitly designed our dataset to necessitate temporal understanding. Approximately 40% of our question-answer pairs fall into the "temporal understanding" category. Besides, we evaluated the performance of tarsier2 with one frame per video sample. As shown in Appendix Table 13, Tarsier2's accuracy drops significantly in a one-frame setting (40.3%) compared to full video (62.8%), confirming the necessity of temporal information.
>
> | Setting | Accuracy |
> | ------- | -------- |
> | text-only | 20.6%     |
> | one-frame | 40.3%     |
> | full-video| 62.8%     |

---

### Official Review · Reviewer_RsGD · 2025-11-01

**Soundness:** 3
**Presentation:** 3
**Contribution:** 3
**Rating:** 6
**Confidence:** 3

**Summary:**

The paper presents a new dataset to evaluate model capability for domain generalization in a video comprehension task. Videos are filtered using a multimodal LLM to ensure they fall into certain domains. The videos are then filtered based on the actions again, using a multimodal LLM so that videos in the different domains have similar semantics. QA pairs are automatically generated for each video, which are then reviewed by human experts. The dataset offers both training and test sets. The paper provides evaluation results over multiple models in multi-domain fine-tuning, single-domain fine-tuning, and zero-shot settings. Human evaluation is also provided to show the relevance and correctness of the automatically generated WA pairs.

**Strengths:**

1. I believe the paper offers a useful dataset (once it is published online), which should contribute to the community.
2. The paper also offers comprehensive details of how the dataset is created.

**Weaknesses:**

1. This may be a minor concern, but since the dataset only offers limited semantics, the evaluation capability may also be limited. It would be nice if the paper could provide some discussion on this point (e.g., whether the set of actions is sufficient for evaluating general performance, and why).
2. As domain generalization can involve fine-tuning, it is desirable to evaluate possible bias in QA pairs and semantics. This may be done by training a language model with only QA pairs and some tokens to represent semantics.
3. It’s hard for me to see what the human evaluation results in Table 10 mean. Human-annotated QAs and some randomization (e.g., randomly pairing up a video and QA in different domains but in the same semantics, or just random video and QA pairs) may help understand it.
4. I also don’t see why human evaluation uses a 5-point scale. Relevance and correctness sound more like binary.

**Questions:**

I want to see some discussion on the weaknesses.

---

> ### Author Response · Authors · 2025-11-21
>
> **1. Regard Limited Semantics**
>
> Thank you for your valuable feedback. We would like to clarify that the core design goal of VUDG is not to cover all possible video semantics, but rather to provide a controlled and fair evaluation platform for domain generalization research in video understanding. This method of carefully selecting semantic categories is a standard practice in domain generalization research; for example, the classic image DG dataset PACS [1] contains only a few categories, aiming to separate the generalization challenges of models facing domain variations from large-scale classification tasks. The action categories in VUDG are carefully selected. We combine data-driven approaches (statistically analyzing high-frequency actions in video sources) and manual screening (supplementing core human daily actions) to ensure that these semantics are representative and consistent across domains. Therefore, we believe that this focused yet diverse set of semantics is sufficient to effectively measure and compare the model's ability to transfer knowledge across different visual domains, thus directly serving our core goal of evaluating domain generalization performance.
>
> Reference:
> [1] D. Li, Y. Yang, Y.-Z. Song, and T. M. Hospedales, “Deeper, broader and artier domain generalization,” in Proceedings of the IEEE International Conference on Computer Vision (ICCV), 2017, pp. 5542–5550.
>
> **2. Regard Language Bias**
>
> |Cartoon|Game|Movie/TV|Virtual|Egocentric|Surveillance|Shaky|Foggy|Night|Rainy|Snowy|Avg|
> |---|---|---|---|---|---|---|---|---|---|---|---|
> |23.3|21.5|20.7|18.3|21.0|22.6|22.8|20.6|20.3|17.8|17.9|20.6|
>
> We agree that this is a critical aspect for a robust benchmark. While training a language-only model is a valid approach, we opted for a more direct and stringent method to assess the inherent language bias in our QA pairs. Specifically, we evaluated a powerful, pre-trained Large Vision-Language Model (Tarsier2-7B) in a zero-shot, text-only setting. We believe this approach more effectively isolates the bias present in the data itself, without introducing confounding variables from a specific training process.
>
> Our results show that when provided with only the question text, the model achieves an average accuracy of just 20.6% across all 11 domains, which is only marginally above the 16.7% random-guess baseline for our six-choice format. The detailed per-domain results, as shown in the table below, further confirm this finding consistently across diverse domains. This low performance strongly indicates that the QA pairs in VUDG do not contain significant linguistic shortcuts and that models must genuinely rely on the visual content to arrive at the correct answer.
>
> **3. Regard Human Evaluation Results**
>
> Thank you for this insightful suggestion. We compared our final annotations against two control groups to contextualize their quality. The first baseline, "random match," directly addresses your suggestion by randomly pairing questions with answers from other QA pairs, serving as a lower bound for question quality. The second baseline, "filtered," consists of QA pairs that were generated but subsequently discarded during the human review phase. As the results indicate, our final VUDG annotations achieved significantly higher scores (4.78 for correctness and 4.62 for relevance) than both the randomly matched pairs (1.23 and 1.45) and the filtered-out pairs (2.36 and 3.58). This large margin clearly validates the high quality and reliability of our dataset's annotations. We will clarify this comparison in the revised manuscript to make the significance of our human evaluation results more explicit.
>
> | |correctness|relevance|
> |---|---|---|
> |random match|1.23|1.45|
> |filtered|2.36|3.58|
> |Ours|4.78|4.62|
>
> **4. Regard Relevance and Correctness**
>
> Thank you for your valuable feedback. When evaluating open-ended answers generated by large visual language models for videos, the quality of the answers often lies across a continuous spectrum rather than a simple binary of right and wrong. For example, an answer might be "partially correct": when the standard answer is "A man in a red T-shirt is playing basketball," the model's answer "A man is exercising" is directionally correct but lacks specificity, while the answer "A man in a blue T-shirt is playing basketball" is incorrect on a key attribute. Binary evaluation cannot distinguish the degree of these subtle errors. Similarly, for relevance, an answer might be completely correct but contain redundant information irrelevant to the question. For example, when asked "What is the person in the video doing?", the answer "He is playing guitar, and there is a painting on the wall in the background" is correct in the latter part but irrelevant to the core of the question. Therefore, using a 5-point scale can more finely quantify these subtle differences, providing a more accurate and informative assessment of the model's performance in domain generalization.

---

### Official Review · Reviewer_M1bM · 2025-11-02

**Soundness:** 2
**Presentation:** 3
**Contribution:** 3
**Rating:** 4
**Confidence:** 4

**Summary:**

This paper introduces the Video Understanding Domain Generalization (VUDG) dataset, a dataset designed to evaluate domain generalization of video understanding models. The authors key motivation is that existing video understanding benchmarks either do not measure domain generalization, or are ineffective at measuring it due to large semantic gap between domains. VUDG aims to address this by collecting videos from 11 distinct domains while keeping the underlying semantic content consistent through a shared set of daily human activities. the authors employ a progressive multi-expert annotation framework that combines multiple large vision-language models for automated question generation, verification, and filtering, followed by a human verification. This process yields a training dataset of 31k QA pairs, and a testing dataset of 4k QA pairs. The authors additionally benchmark existing video understanding models on VUDG.

**Strengths:**

The proposed benchmark is the first large-scale benchmark designed to measure domain generalization in VLMs, consisting of a dedicated training/testing set and clear protocols for evaluation

The choice of domains and corresponding videos is high quality: The domains selected in VUDG are broad and diverse and a large portion of the videos used for testing are newly collected by the authors, reducing the risk of data leakage with existing VLM training data

**Weaknesses:**

The reviewer understands the primary motivation behind VUDG is to unify the semantic space across domains, and this intuitively makes sense to the reviewer. However, it is not scientifically shown why existing benchmarks fail to assess domain generalization due to the lack of cross-domain semantic alignment
* What evidence exists to motivate the community to evaluate the domain generalization ability of their models with VUDG, instead of a benchmark like Video-MME that also contains multiple video domains?
* The t-SNE in Figure 6 does show the semantic alignment between domains in VUDG, but it is not compared with other benchmarks

It is unclear if the testing dataset exhibits language or visual bias in the MCQ questions, which is a common pitfall of LLM generated MCQs (Zohar et al., Apollo: An Exploration of Video Understanding in Large Multimodal Models, CVPR 2025). The authors mention there is a human verification step, but do not explicitly state that it addresses this bias.
* Additionally there are no qualitative examples of MCQs in the paper. As this is a dataset paper the reviewer expected to see many more examples of QA pairs in the dataset

The choice of VLMs used for evaluation in the full fine-tuning and single-domain generalization settings is relatively shallow, reducing the insight gained from these settings

**Questions:**

What evidence exists to motivate the community to evaluate the domain generalization ability of their models with VUDG, instead of a benchmark like Video-MME that also contains multiple video domains? Can this semantic shift be quantified and shown to hurt a benchmarks ability to assess domain generalization? Or can it be qualitatively shown?

How can the authors be certain that the MCQs in VUDG do not suffer from visual or language bias? Are there any failsafes in the data curation to mitigate this?

---

> ### Author Response · Authors · 2025-11-21
>
> **1. Regard Semantic Consistency Problem**
>
> Thank you for your valuable feedback. According to the classic definition of domain generalization [1], its goal is to train a model on one or multiple source domains and it can generalize to unknown target domains with consistent semantic space but different data distributions (domain shifts). This allows us to test whether the model has learned essential, generalizable knowledge, rather than overfitting to the surface features of the source domains (such as style or background).
>
> Existing benchmarks like Video-MME, while encompassing multiple video domains, exhibit significant differences in semantic content between different domains (e.g., sports, art). Therefore, when models are evaluated across domains on these benchmarks, it's difficult to determine whether the performance degradation stems from an inability to handle domain shifts or from a lack of semantic knowledge across these different domains.
>
> VUDG is designed to address this evaluation confusion. By constraining cross-domain semantic consistency, we ensure that inter-domain differences primarily arise from non-semantic variations in style, environment, or perspective.
>
> To provide direct evidence for this, as you suggested, we have added a t-SNE visualization of the question embeddings from Video-MME to the appendix (Section D, Figure 9). This visualization clearly shows that questions from specific domains, such as 'Sports Competition' and 'Artistic Performance', form distinct and isolated semantic clusters. This starkly contrasts with the significant cross-domain semantic overlap demonstrated in VUDG. Therefore, VUDG is specifically designed to truly and purely measure a model’s robustness to understanding core semantic content in the face of distributional variations, thus providing the community with a standardized platform to evaluate real-world “domain generalization” capabilities, rather than broad “generalization” capabilities.
>
> Reference:
> [1] K. Zhou, Z. Liu, Y. Qiao, T. Xiang and C. C. Loy, "Domain Generalization: A Survey," in IEEE Transactions on Pattern Analysis and Machine Intelligence, vol. 45, no. 4, pp. 4396-4415, 1 April 2023, doi: 10.1109/TPAMI.2022.3195549.
>
> **2. Regard Language or Visual bias**
>
> How to avoid the potential language and visual biases of generated data is a fundamental and widely faced problem, and we indeed consider this problem in the VUDG dataset construction. To ensure the robustness of our MCQs, we implemented several safeguards in our data curation pipeline. (1) To address the issue of language bias, we removed some obvious questions that could be deduced through common sense during the human review phase. (e.g., Q: What color is the Christmas hat that all the characters are wearing? A: The hat is red and white.) To quantitatively demonstrate this, we conducted a "text-only" baseline experiment where the Tarsier2-7B answered questions in VUDG using only the QA pairs without any video input. The resulting accuracy was merely 20.6%, only slightly above the random guess rate of 16.7%, which indicates that models cannot easily infer the correct answer from the text alone. (2) As for visual bias, we proactively mitigated this issue during the question design phase by intentionally including a significant number of questions that necessitate temporal understanding, such as "What is the correct order of events in the video?". Besides, we evaluated the performance of tarsier2 with one frame per video. As shown in Table 13 of the Appendix, Tarsier2 achieves much lower accuracy in one-frame setting than using full video, indicating the critical role of temporal dynamics information in VUDG.
>
> |Setting|Accuracy|
> |---|---|
> |text-only|20.6%|
> |one-frame|40.3%|
> |full-video|62.8%|
>
> **3. Regard Qualitative Examples**
>
> We sincerely thank the reviewer for this valuable suggestion. To address this, we have revised the manuscript and included a new section (Section I) in the Appendix, where we now showcase three detailed examples of question-answer (QA) pairs, complete with their multiple-choice options, for each of the four distinct question types in our dataset.
> These examples demonstrate that QA pairs in our dataset generally struggle to derive answers from questions, and most questions rely on temporal understanding or localization.
>
> **4. Regard Lack of Finetuning Models**
>
> We sincerely appreciate the reviewer’s suggestion to expand the selection of VLMs for the full fine-tuning and single-domain generalization settings. Our initial selection was constrained by the significant computational resources required. However, we fully recognize that including a broader range of models will provide deeper insights into the robustness of current methods. Therefore, we are actively conducting additional experiments with more diverse and state-of-the-art models during the rebuttal period and will incorporate these results to ensure a more comprehensive evaluation in the final version.

---

### Official Review · Reviewer_xqBB · 2025-11-03

**Soundness:** 3
**Presentation:** 3
**Contribution:** 3
**Rating:** 6
**Confidence:** 2

**Summary:**

Authors introduce VUDG, a video-understanding domain generalization benchmark spanning 11 domains with a shared semantic space. QA pairs are built via a multi-expert (model-assisted + human) annotation pipeline. Across multi-source, single-source, and zero-shot protocols, both classic VideoQA methods and state-of-the-art LVLMs show sizable accuracy drops under domain shifts, revealing uneven robustness and a clear gap from in-domain fine-tuning.

**Strengths:**

Innovative Benchmark Design

The paper introduces a novel benchmark specifically crafted to test how well video understanding models generalize across domain shifts, filling an important gap in current evaluation practices.

Thorough and Systematic Evaluation

It provides a comprehensive assessment of multiple domain shift scenarios (style, viewpoint, weather, lighting) using both traditional VideoQA models and modern LVLMs, offering a well-rounded perspective on robustness.

**Weaknesses:**

Insufficient Distractor Analysis

While the paper reports aggregate counts and durations (Fig. 4–5), it provides no fine-grained QA/distractor analyses - e.g., answer-length distributions, length-matching of options, distractor similarity, so on. For example, the prompt states to have distractors be the same length but from my experience LLMs might just ignore it completely and there is no statistics to verify if it is the case. Further investigation into questions and answer options would strengthen the benchmark’s validity

LLM dependence and possible circularity

The pipeline relies on LLMs for generation, screening, and even as an automated judge for open-ended scoring, which can introduce model bias and evaluation circularity despite a final human pass.

**Questions:**

refer to weaknesses

---

> ### Author Response · Authors · 2025-11-21
>
> **1. Regard Insufficient Distractor Analysis**
>
> We thank the reviewer for this insightful suggestion. We have added a detailed statistical analysis of the answer options for the validity and quality of our VUDG benchmark.
>
> Our analysis confirms that the options are well-balanced and the distractors are semantically meaningful. Specifically, (1) the average word count for correct answers (5.85) and incorrect distractors (5.55) is highly similar, mitigating potential length-based shortcuts for models. (2) To evaluate the quality of our distractors, we computed the cosine similarities between the embeddings of the correct answer and other options. The average similarity between the correct answer and curated distractors is 0.720, which is significantly higher than the average similarity of randomly sampled answers from another QA pairs (0.461). This indicates that our distractors are semantically relevant to the correct answer and pose a genuine challenge. (3) We find that there is less positional bias in options of correct answers, as shown by their uniform distribution across all possible positions.
>
> | Option Position | A | B | C | D | E | F |
> | -------- | -------- | -------- | -------- | -------- | -------- | -------- |
> | Count    | 6202     | 6069     | 5861     | 6061     | 6068     | 6127     |
>
> These statistics collectively demonstrate that VUDG is a carefully constructed and robust benchmark. We will incorporate these detailed analyses into the revised version of our paper to further strengthen its contribution.
>
> **2. Regard LLM dependence and possible circularity**
>
> Thanks for your valuable comment. How to avoid the model biases and evaluation circularity of generated data introduced by large language models is a fundamental and widely faced problem, and we indeed consider this critical problem in the VUDG construction, that is the designed multi-expert progressive annotation framework. Specifically, (1) to break the potential for evaluation circularity from a single model's inherent biases, we deliberately employed three distinct large models (Gemini 2.5 Flash, DeepSeek V3, and Gemini 2.5 Pro) at different key stages: open-ended QA generation, multiple-choice option generation, and review. (2) To control model biases at the source, we constrained the generation scope by pre-defining 37 activity scenes during data collection and utilized meticulously designed prompts. For instance, in attribute recognition tasks, our prompts explicitly guided the models to focus on verifiable and objective attributes such as shape, position, motion, and state. (3) In the human study, our dataset achieved correctness and relevance scores of 4.78 and 4.62 in open-end QA, significantly higher than those of the QA pairs filtered out during the human review phase (2.36 for correctness and 3.58 for relevance), validating the effectiveness of our annotation framework.

---

### Author Response · Authors · 2025-12-01
**General Response**

We sincerely thank all reviewers for their thoughtful comments, insightful critiques, and constructive suggestions. These have greatly helped us clarify the motivation, methodology, and contributions of our work.

Reviewers consistently highlighted the following strengths:

* First large-scale benchmark for domain generalization in video understanding, featuring diverse domain shifts (e.g., style, viewpoint, weather, lighting).
* High-quality test set largely composed of newly collected videos, minimizing data leakage.
* Thorough evaluation across both traditional VideoQA models and modern LVLMs.
* Thoughtful design: including temporal reasoning in questions and a scalable LLM-based annotation pipeline, enhancing practical value and community impact.

Below, we summarize key points that address concerns raised across reviews.

**1. The Significance of Video Understanding Domain Generalization**

Although large-scale pre-training has significantly improved zero-shot generalization, reliance on massive, often proprietary datasets (e.g., 40 million videos in models like Tarsier2) remains impractical for most researchers due to computational cost and data accessibility constraints. Moreover, many real-world deployments require lightweight, efficient models that must generalize well from limited domain-specific data.
Crucially, even powerful models such as Tarsier2 or closed-source models like GPT-4o do not achieve state-of-the-art performance on VUDG, highlighting that scaling alone does not solve the core challenges of video understanding DG. Thus, domain generalization in video understanding remains a fundamental and enduring problem.

**2. Difference Between Video DG and Image DG**

While domain generalization in images has been extensively studied, video understanding introduces unique challenges rooted in the temporal dimension. Video understanding domain generalization needs to address temporal domain shifts, including changes in action speed, camera motion (e.g., shaky footage), or action rhythm, which have no counterpart in static images. To address this, our benchmark VUDG is explicitly designed to evaluate model robustness against various domain shifts, offering the community the first controlled and comprehensive setting for video understanding DG evaluation.

**3. Language and Visual (Static) Bias**

We rigorously evaluated potential shortcuts in our multiple-choice QA design. In a text-only setting, even a strong VLM (Tarsier2-7B) achieves only 20.6% accuracy (vs. 16.7% random chance), confirming minimal language bias. Moreover, performance drops significantly when using only one frame (40.3% vs. 62.8% with full video), demonstrating that VUDG questions genuinely require temporal reasoning, not static visual cues.

**4. Cross-Domain Semantic Consistency**

A core principle of DG is that source and target domains share the same semantic label space, differing only in data distribution. In contrast, existing benchmarks like Video-MME include semantically distinct domains (e.g., sports vs. art) and are not suitable for evaluating domain generalization ability. To validate our design, we include a t-SNE visualization (Appendix D, Fig. 9) showing that VUDG exhibits strong cross-domain semantic overlap, while Video-MME forms isolated clusters. This ensures VUDG isolates the DG challenge from confounding semantic inconsistency.

**5. Examples of Multiple-Choice QA Pairs**

In response to requests for qualitative examples, we have added detailed multi-choice QA samples for all four question types in Appendix Section I. These illustrate that answers cannot be guessed from the question alone and often require temporal localization or reasoning.

**6. Adding Baselines for Human Study**

We compared the final annotations with two newly added baselines in human study: (i) randomly matched QA pairs and (ii) generated pairs filtered out during the review phase. Our dataset achieved 4.78/5 (accuracy) and 4.62/5 (correlation), significantly better than the two baselines, confirming high annotation quality and correlation.

**Revised Manuscript:**
* Provide a more detailed discussion of static (language and visual) bias (see Appendix Section C)
* Add a t-SNE visualization comparing cross-domain embeddings with Video-MME (see Appendix Section D)
* Include qualitative example of multiple-choice QA pairs (see Appendix Section I)
* Expand the human study to include two baselines (see Section 4.5)
* Include zero-shot results of Tarsier2 and GPT-4o on VUDG (see Section 4.3)


We believe these clarifications and additions strengthen the rigor, transparency, and impact of VUDG. We are committed to incorporating all feedback to improve the final version of the paper and hope the reviewers find our responses satisfactory.

---

### Meta-Review · Area_Chair_RJjj · 2026-01-05

**Summary:**

The paper introduces a dataset/benchmark for evaluating domain generalization in video understanding tasks. The dataset addresses 11 domains covering a variety of sources of distribution shift while maintaining semantic consistency between domains. The annotation pipelines provides a structured VideoQA. The paper justifies the need clearly (there is a clear need for such a benchmark), the design of the dataset (including the QA piece) makes it interesting and useful to a broader class of problems. A key concern was LLM dependence and circularity, though the authors have addressed this. The reviewers provide several recommendations that are not severe limitations, but will help improve the paper in a camera-ready version if possible.

**Reviewer Concerns:**

The authors have satisfactorily addressed much of the critical feedback.

**Reviewer Scores:**

I conjecture the scores would go up on average by a point had there been a chance for a full-discussion.

---

### Decision · Program_Chairs · 2026-01-26

Accept (Poster)